# Temperature Effect on the Corrosion Inhibition of Carbon Steel by Polymeric Ionic Liquids in Acid Medium

**DOI:** 10.3390/ijms24076291

**Published:** 2023-03-27

**Authors:** Giselle Gómez-Sánchez, Octavio Olivares-Xometl, Paulina Arellanes-Lozada, Natalya V. Likhanova, Irina V. Lijanova, Janette Arriola-Morales, Víctor Díaz-Jiménez, Josué López-Rodríguez

**Affiliations:** 1Facultad de Ingeniería Química, Benemérita Universidad Autónoma de Puebla, Av. San Claudio y 18 Sur, Ciudad Universitaria, Col. Jardines de San Manuel, Puebla 72570, Mexico; giselle.gomezsan@alumno.buap.mx (G.G.-S.); oxoctavio@yahoo.com.mx (O.O.-X.); janette.arriola@correo.buap.mx (J.A.-M.); victor.diazj@alumno.buap.mx (V.D.-J.); 2Gerencia de Materiales y Desarrollo de Productos Químicos, Instituto Mexicano del Petróleo, Eje Central Lázaro Cárdenas No. 152, Col. San Bartolo Atepehuacan, Ciudad de México 07730, Mexico; nvictoro@imp.mx; 3CIITEC, Instituto Politécnico Nacional, Cerrada Cecati S/N, Colonia Santa Catarina, Azcapotzalco, Ciudad de México 02250, Mexico; irinalijanova@yahoo.com.mx; 4ESIQIE, Instituto Politécnico Nacional, Zacatenco, Av. Luis Enrique Erro S/N, Nueva Industrial Vallejo, Gustavo A. Madero, Ciudad de México 07738, Mexico; jjlopezr@ipn.mx

**Keywords:** vinylimidazolium, API 5L X60 steel, mixed-type inhibitors, EIS, polarization, SEM, DFT

## Abstract

In the present research work, the temperature effect on the corrosion inhibition process of API 5L X60 steel in 1 M H_2_SO_4_ by employing three vinylimidazolium poly(ionic liquid)s (PILs) was studied by means of electrochemical techniques, surface analysis and computational simulation. The results revealed that the maximal inhibition efficiency (75%) was achieved by Poly[VIMC4][Im] at 308 K and 175 ppm. The PILs showed *E_corr_* displacements with respect to the blank from −14 mV to −31 mV, which revealed the behavior of mixed-type corrosion inhibitors (CIs). The steel micrographs, in the presence and absence of PILs, showed less surface damage in the presence of PILs, thus confirming their inhibiting effect. The computational studies of the molecular orbitals and molecular electrostatic potential of the monomers suggested that the formation of a protecting film could be mainly due to the nitrogen and oxygen heteroatoms present in each structure.

## 1. Introduction

The use of corrosion inhibitors (CIs) is a frequent practice for preventing the corrosion phenomenon from occurring, which is based on the addition of chemical substances at low concentration to the corrosive medium [1]. CIs are employed internally in pipeline systems and carbon steel containers [2] as a low-cost-corrosion-control alternative whose action mechanism proceeds through the adsorption of CI molecules on metal surfaces, thus diminishing the corrosion rate of the metallic system to be protected; industry sectors such as the exploration and production of oil and gas, oil refineries, production of chemical products, heavy industry, water treatment and product additive industry are normally benefited by the protective action of CIs [3]. The main advantage offered by CIs before other control methods is that their implementation does not require a process stop [4,5]. CIs are divided into two big classes: inorganic and organic. Anodic inorganic CIs include nitrates, chromates, molybdates and phosphates, whereas cathodic ones are represented by zinc derivatives and polyphosphates [6,7]. In contrast, organic CIs are mainly film-forming compounds that work through physical and/or chemical adsorption processes [8]; in general, these are compounds with heteroatoms (P, S, N and O) and π bonds that include amines, amides, imidazolines, sodium benzoate mercaptans, esters and ammonia derivatives [6]. The performance of a CI is based on a competition process between CI molecules and corrosive ions (*H^+^*, *H*_3_*O^+^*, *Cl^−^*, and *SO*_4_^2−^, among others) to occupy active sites on a metallic surface, where the higher the number of CI adsorbed molecules, the better the inhibition efficiency (*IE*) against the corrosive medium in contact with the metallic material [9]. The inhibition process is affected by many factors that modify the stability of CIs in corrosive media such as the type of metallic surface, temperature, immersion type, medium flow rate, pH, medium ion concentration and water hardness, among others [10]. It is well known that the *IE* of most CIs diminishes with the temperature increase due to a diminution of the adsorption strength on a metallic surface, thus generating a desorption process; then, the performance of an inhibitor depends mainly on the temperature [11]. For this reason, polymeric CIs are a viable alternative in corrosive processes at temperatures above 298 K, for they feature diverse functional groups in one molecule that can form complexes with metallic ions and occupy a higher surface area, thus protecting the metallic material from corrosive agents. From this type of compounds, the following have been the most studied: carbohydrates, polysaccharides, polysulfide, phosphate esters, polycarboxylates/polycarboxylic acids, polyanilines, polyaspartates and other polyaminoacids, and polyvinylamide and polyamine derivatives [2]. Their inhibiting behavior is structurally reinforced by the presence of cyclic rings, double and triple bonds and heteroatoms such as oxygen and nitrogen that work as adsorption active centers [12]. It has been reported that the protection against corrosion by polymeric CIs has been above 90% in acid, sour and sweet media [13]. The *IE* and stability of the polymer protecting film highly depend on the hydrophilic and hydrophobic features of the polymers, where a suitable relationship is fundamental for the inhibition activity to be satisfactory [14].

Poly(ionic liquid)s (PILs) are a special group of polymeric compounds [15,16] whose design is wide and involves a big variety of monomers that provide unique properties for specific applications [15,17]. PILs possess a macromolecular structure that does not only consist of a polymeric skeleton, but also of at least of an ionic liquid (IL) monomer, which along with other polymeric species produces additional properties to those of a conventional IL such as stability in aqueous media, the presence of polymeric chains that can displace a higher number of water molecules from the metallic surface, mechanical stability, autoassembling and, above all, the presence of multiple adsorption centers that contribute to a slower desorption process that favors the formation of complexes with the metallic surface [18]. The presence of an IL monomeric species in the structure promotes two processes that reinforce each other through a synergistic effect: (a) the charge transfer process from the IL functional groups to the metallic *d* orbital, “donation” and (b) the intraelectronic repulsion process through which the metal transfers its electrons to empty IL orbitals, “backdonation” [14]. Notwithstanding, despite the aforementioned properties, their application as CIs has not been studied enough. Wang et al. [19] and Odewunmi et al. [20] carried out an interesting comparison between the structure of ILs and their corresponding polymeric forms as PILs to be potentially used as CIs of steel in HCl and found that PILs displayed a better inhibiting effect than their related IL species alone due to a higher number of functional groups in their structure. Additionally, Odewunmi et al. stated that PILs with halide anions provide an excess of electrons that allows for the attraction and adsorption of the polymeric cationic species.

Table 1 shows some studies performed with polymeric CIs and their behavior as a function of temperature. Different types of polymers have been evaluated as CIs at different temperatures: PILs [19,20], synthetic polymers [21], biomacromolecules [22], carbohydrates [23,24], recycled PET oligomers [25] and triblock copolymers [26,27]. The temperature effect of polymeric CIs has been analyzed by different authors such as Alaoui et al. [21], who suggested that the *IE* behavior with respect to the medium temperature is associated with the adsorption type change that presents a polymer; in this way, the increase in the *IE* is related to a chemisorption process that is favored by a temperature increase. In contrast, Gowraraju et al. [23], Charitha et al. [24] and Yasir et al. [25] observed that an *IE* decrease as the temperature increases can be explained by a fast adsorption-desorption process and by the decomposition and/or reordering of inhibiting molecules led by a prevailing physisorption process. An implicit variable in the temperature effect on corrosion inhibition processes is the activation energy (*E_a_*). Chauhan et al. [22], Yasir et al. [25] and Kumar et al. [26,27] stated that a high *E_a_* value in the presence of a CI suggests the formation of a physical barrier attributed to the interaction between inhibiting molecules and the active sites of a metallic surface promoted by the adsorption process of heteroatoms, cyclic rings and functional groups that increase the thickness of the electrical double layer for the formation of an inhibiting film that diminishes the metal corrosion rate.

In a previous study carried out by the authors of the present work, three vinylimidazolium-derived PILs were evaluated as CIs of API 5L X60 steel in 1 M H_2_SO_4_ at 298 K, concluding that such PILs behaved effectively as CIs [28]. For this reason, in this manuscript, the temperature effect on the corrosion inhibition process of API 5L X60 steel employing three PILs was investigated by means of polarization and impedance electrochemical analysis, mass loss, surface analysis and computational calculations.

## 2. Results and Discussion

### 2.1. Weight Loss

The mass loss test is a frequently employed method for evaluating the performance of CIs and establishing the corrosion rate (*V_corr_*) of a metallic material under different conditions [29]. The *V_corr/_*Δ*W* ratio as a function of the inhibitor concentration (*C_INH_*) at 308 K is shown in Figure 1, where the increase in concentration provoked less metallic dissolution and then lower *V_corr_*; similar results were obtained with the other evaluated temperature values.

Table 2 presents the steel *V_corr_* values at different temperatures and *C_INH_* after 4 h of immersion in 1 M H_2_SO_4_. It can be observed that the temperature produced an increase in *V_corr_*, even in the presence of PILs, which was attributed to desorption phenomena of the inhibiting macromolecules. In addition, it is shown that the lowest *V_corr_* values were obtained at *C_INH_* of 175 ppm, indicating that a higher amount of PIL macromolecules can form a more homogeneous protecting film that reduces the diffusion of sulfate ions toward the metallic surface [30,31,32].

### 2.2. Electrochemical Measurements

Figure 2 shows the *E_OCP_* behavior as a function of the immersion time of the metallic sample in 1 M H_2_SO_4_ in the absence and presence of PILs at different temperatures. In the presence of PILs, it can be observed that the *E_OCP_* curves diminish toward more negative values with respect to the blank at the different evaluated temperatures. The *E_OCP_* displacement intervals at 308, 318 and 328 K ranged from −439 to −453 mV, from −420 to −452 and from −420 to −451 mV, respectively. This behavior pattern is associated with a fast PIL inhibitor adsorption due to the formation of a protecting film on the metallic surface. On the other hand, the diminution of the blank *E_OCP_* with the temperature change involved higher metallic degradation. Furthermore, it can be seen that in all the systems, the *E_OCP_* stability was reached at 600 s of immersion, approximately.

The inhibition behavior of the PILs assessed by the LPR and PDP tests at different temperatures and concentrations is shown in Figure 3, Figure 4 and Figure 5. The addition of Poly[VIMC4][Im], Poly[VIMC2][Br] and Poly[VIMC4][Br] to the corrosive medium modified the slopes of the steel overpotential (*η*)—current density (*i*) lines, where a lower slope is related to higher resistance to polarization (*Rp*) in the metallic interface. As for the PDP results, it is observed that the addition of CIs provoked the diminution of *i* and the potential displacement (Δ*E_corr_*) toward more negative values. For all the systems, *i* was a function of the *C_INH_* of PILs, where the Tafel curves displayed a lower *i* at 175 ppm.

Table 3 shows LPR and PDP electrochemical parameters obtained from Figure 3, Figure 4 and Figure 5. The changes of the *Rp* values of the CIs with respect to the blank (Δ*Rp = Rp_CI_ − Rp_Blank_*) were 30.5, 8.2 and 2.0 Ω cm^2^ at 308, 318 and 328 K, respectively, whereas the corrosion current density (*i_corr_*) changes, in the same order (Δ*i_corr_ = i_corr,Blank_ − i_corr,CI_*), were 917, 1441 and 2898 µA cm^−2^. This behavior pattern is attributed to the adsorption of the CIs through the blocking of the metallic surface active sites [33]. From Table 3, it can be concluded that the temperature increase accelerated the dissolution of the metallic surface in the presence of H_2_SO_4_ and the different PILs [34]. As for Δ*E_corr_*, displacements from −14 mV to −31 mV are displayed within the ±85 mV interval and indicate that the PILs exhibited the behavior of mixed-type CIs [35]; the Δ*E_corr_* trend toward more negative values suggests higher activity in the cationic part *[Im^+^]* of the PILs. Furthermore, their addition to the corrosive medium with respect to the systems without CI provoked changes in the Tafel cathodic slopes (*β_C_*), thus confirming that the presence of Poly[VIMC4][Im], Poly[VIMC2][Br] and Poly[VIMC4][Br] affected the iron dissolution reactions, but mainly the evolution of H_2_ due to the adsorption of the PILs on the predominantly cathodic active sites, which retarded the electron transfer process necessary for the generation of hydrogen [36,37].

The *IE*s of the PILs obtained at 308, 318 and 328 K are shown in Figure 6. The results display maximal efficiencies at 175 ppm. The *IEs* exhibited the following trend: Poly[VIMC4][Im] > Poly[VIMC4][Br] > Poly[VIMC2][Br]. The diminution of the *IEs* with the *T* increase implies higher kinetic energy in the redox reactions and, as a consequence, the increase in the steel anodic dissolution, which provoked the desorption of the PIL molecules and less covered fraction [34,38]. The *IEs* of Poly[VIMC4][Br] and Poly[VIMC2][Br] diminished in ~33% with the temperature increase, which was related to the low contribution of the non-ionic blocks (acrylamide and vinylpyrrolidone) to the adsorption process. As for Poly[VIMC4][Im], the *IE* diminished in ~16%, indicating a higher PIL stability before the *T* increase; this result is associated with the presence of the imidazolate monomers of butyl vinylimidazolium, which along with their reticulated polymeric structure could occupy a higher surface area and block the attack of corrosive ions. The preferentially cathodic Δ*E_corr_* of the PILs would confirm the influence of the cationic part on their inhibition process through the adsorption of vinylalkylimidazolium ions, mainly.

The EIS spectra of API 5L X60 steel in the absence and presence of CIs at 308, 318 and 328 K are shown in Figure 7, Figure 8 and Figure 9, respectively. The behavior of the real (*Z*′) and imaginary (*Z*″) impedances in the Nyquist spectra shows a capacitive depreciated loop controlled by a charge transfer process and an inductive loop at low frequencies. On the other hand, in the presence of PILs, the Bode plots exhibit the displacement of the impedance module *|Z|* toward higher values with respect to the blank, which is associated with the adsorption of the CIs on the metallic surface [39]. The latter suggests that the evaluated CIs retarded the kinetics of the redox reactions part of the corrosion process of steel in acid medium [40]. At intermediate frequencies, the Bode plots presented maximal phase angle values ascribed to the capacitive behavior of the electric double layer in the metal-solution interface. Although the temperature increase diminished the maximal phase angle values, with the addition of PILs, they are higher than those displayed by the blank, which indicates that the protection of the metallic surface by the effect of the PILs prevailed even with the temperature increase [33,41].

The obtained impedance spectra of API 5L X60 steel were fitted by means of the EEC model shown in Figure 10. The result of fitting the EIS experimental data is reported in Table 4. Different electrical elements are described as follows: *R_s_* is the resistive element of the solution, which describes the resistance of the WE before the electrolytic solution. The *R_ct_* and *CPE_dl_* elements represent the charge transfer process in the metal-solution interface. *R_ct_* is the resistance to the charge transfer and *CPE_dl_* is the constant phase element associated with the electric double layer. *R_L_* and *L* are inductive elements that are related to relaxation processes of intermediate species in the oxidation reaction like adsorbed species from the acid medium such as *H*_3_*O^+^* and *SO*_4_^2−^. Finally, *R_f_* and *CPE_f_* are the resistance and constant phase element, respectively, ascribed to a film formed on the metallic surface with different dielectric properties.

The constant phase elements (CPEs) indicate the ideality deviation of the capacitances of the EIS spectra and are represented in the Nyquist plots as depreciated semicircles (Figure 7, Figure 8 and Figure 9). The CPEs were calculated from two parameters: the proportional factor (*Y*_0_) and the exponent *n*. The latter is associated with possible surface irregularities due to roughness, inhibitor adsorption or the formation of porous layers. The CPE impedance is defined by Equation (1):(1)$$Z_{CPE}=Y_{0}{}^{-1}{\left(j\omega \right)}^{-n}$$where *j* is an imaginary number (−1)^1/2^ and *ω =* 2*πf* is the angular frequency of the maximal value of the real impedance; *n* falls within a close interval (−1 *≤ n ≥* 1), where −1, 0 and 1 are usually related to an inductor, resistor and capacitor, respectively [42,43]. The pseudocapacitance derived from a CPE can be calculated by means of Equation (2):(2)$$C={\left(Y_{0}R^{1-n}\right)}^{1/n}$$

In the interfacial phenomena controlled by diffusion, relaxation processes occur at specific frequencies and temperatures; in an electrochemical system, the characteristic constant of such a process within the time domain is known as the relaxation time (*τ_dl_*), which is defined as the time necessary for the charge distribution to recover the equilibrium state and is commonly employed to distinguish the polarization effects that normally overlap in the frequency domain and that can be attributed to underlying physical processes [44,45]. The relaxation time (*τ_dl_*) is given by Equation (3) [46,47]:(3)$$\tau_{dl}=C_{dl}R_{ct}=\frac{1}{2\pi f_{\max}}$$

Table 4 shows the values of the EEC electric elements. It can be observed that the *R_ct_* values are a function of *C_INH_*, where the highest results occurred at 175 ppm. The polarization resistance exhibited by the system by the EIS (*Rp_EIS_*) technique involves all the EEC resistive elements [28]. The *Rp_EIS_* values in the presence of the PILs were higher than those shown by the blank at the different temperatures, indicating higher resistance to the electron transfer in the electrochemical reactions involved in the corrosion process and confirming the inhibiting behavior of the tested compounds. *R_s_* did not display any significant change, which revealed that the corrosive systems underwent a minimal ohmic drop.

The effect observed with the temperature was as follows: the *R_ct_* diminution and increase in the capacitance of the electric double layer (*C_dl_)* reveal a small surface fraction covered by the CIs, which can be attributed to the growing diffusion rate of corrosive ions such as [*H*_3_*O^+^*] and [*SO*_4_^2−^] that promotes the transfer of electrons and steel dissolution. The inductance elements *R_L_* and *L* diminished with the increasing temperature, which suggests that the desorption of species adsorbed on the surface originated mainly by the relaxation process of intermediate compounds involved in oxidation reactions [40]. The reduction of *R_f_* and slight increase in the film capacitance (*C_f_*) shows the possible formation of a film consisting of corrosion products and/or CI adsorbed species that could be removed from the metallic surface more easily by the temperature effect.

The *τ_dl_* values were higher in the presence of PILs with the increasing *C_INH_* (Table 4), revealing that the electric charge and discharge process occurring in the metal-solution interface is slower due to the presence of a higher number of macromolecules adsorbed on the metallic surface [46,48]. However, the number of these molecules falls with the increasing temperature, for it reduces the necessary time for their right orientation; as the temperature rises, the molecular thermal movement increases and *τ_dl_* goes on diminishing, which negatively affects the migration of PIL inhibiting species toward the metallic surface [49].

In summary, at 308, 318 and 328 K, the reduction of the values of the EEC resistive elements was observed, which was associated with the desorption of PIL species forming the inhibiting film on the surface as a result of the intensification of the charge transfer process related to the kinetics of the redox reactions that occurred in the metal-solution interface when the temperature was raised, thus causing a higher anodic dissolution of the metallic sample.

### 2.3. Thermodynamic and Kinetic Properties

The action of a film-forming CI is based on a solid-liquid adsorption process that consists in the union of molecules and/or ions on a metallic surface through either chemical or physical interactions. The degree of surface coverage (θ) as a function of the CI concentration can be represented with adsorption isotherm models such as Langmuir, Temkin, Frumkin, Freundlich and Flory-Huggins, among others [50].

Figure 11 shows the adsorption process of the PILs Poly[VIMC4][Im], Poly[VIMC2][Br] and Poly[VIMC4][Br] on the steel surface at 308 and 318 K, which can be described by the Langmuir adsorption isotherm (Equation (4)) due to the correlation coefficients (R^2^) close to unity. This model is related to ideal adsorption with the formation of a monolayer on a finite number of identical and equivalent adsorption sites, considering that there is no lateral interaction between the adsorbed molecules [51,52]:(4)$$K_{ads}C_{inh}=\frac{\theta }{\left(\theta -1\right)}$$

By increasing the temperature to 328 K, a better fitting of the experimental data was obtained with the Freundlich isotherm (Figure 11) whose model is described by Equation 5. This model is associated with non-ideal, reversible and heterogeneous adsorption [53]. This change in the adsorption isotherm model is ascribed to a desorption process of the CIs with the increasing temperature, which generates more unprotected metallic sites by the attack in the corrosive medium. The thermodynamic parameters obtained for each adsorption isotherm at different temperatures are shown in Table 5.
(5)$$K_{ads}C_{inh}=\theta$$

It is known that the adsorption equilibrium constant (*K_ads_*) represents either the adsorption or desorption strength between the adsorbate (PILs) and adsorbent species (API 5L X60 steel), which indicates the equilibrium relationship between the CI concentration on the metallic surface and in the solution core [54]. From the linear regression of the plots displayed in Figure 11, the *K_ads_* values were estimated (Table 5). High *K_ads_* values reveal a better adsorption efficiency of the CIs and suggest that the adsorption was favored by forming a stable film on the steel surface [43,55], notwithstanding, the temperature increase promoted the diminution of *K_ads_*, which showed that the interaction between the PILs and steel surface was weakened and, as a consequence, the adsorbed species could be desorbed at higher temperatures [56].

With the *K_ads_* values, the change in the standard Gibbs free energy of adsorption (Δ*G°_ads_*) was obtained by means of Equation (6) [57]:(6)$$\Delta G{{}^{\circ}}_{ads}=-RTln\left(55.5K_{ads}\right)$$

The negative values of Δ*G°_ads_* express that the adsorption was thermodynamically spontaneous [9]. The values of Δ*G°_ads_* are associated with the type of adsorption that occurred in a metal-liquid system [58,59,60]: when Δ*G°_ads_* > −20 kJmol^−1^, physical adsorption or physisorption took place, which is characterized by the electrostatic interaction between the charge of the CI molecules and the charge of the metallic surface; when Δ*G°_ads_* < −40 kJmol^−1^, the behavior indicates that the CI molecules and metallic surface share or transfer their charge in such a way that a coordinate metallic bond is formed, having chemical adsorption or chemisorption. When Δ*G°_ads_* displays intermediate values between −20 and −40 kJmol^−1^, it is considered that the CI undergoes an adsorption process that is both physical and chemical.

The obtained Δ*G°_ads_* values (Table 5) of the PILs evaluated at 308, 318 and 328 K ranged from −26 to −35 kJmol^−1^, which suggested a physicochemical adsorption process, where adsorption started under electrostatic forces (physisorption) between the steel surface and functional groups of the PIL polymeric blocks. Furthermore, the formation of coordinate complexes between the substituents of the mentioned PILs and vacant *d* iron orbital of the metallic surface was possible [61,62]. However, the chemisorption process had a minor contribution to the adsorption mechanism of the PILs at the different evaluated temperatures [59,63]. The slight growth in the Δ*G°_ads_* values as a consequence of the system temperature increase suggested that the desorption process of the PIL adsorbed species was favorable, making their inhibition process difficult and promoting the anodic dissolution of iron.

In order to understand the behavior of the PILs with respect to the temperature, the standard adsorption enthalpy (∆*H°_ads_*) was calculated from the Van’t Hoff model, as indicated in Equation 7 [18,64]. ∆*H°_ads_* stems from the slope obtained by linear regression of ln*K_ads_* vs. 1/*T*, whereas the standard adsorption entropy (∆*S°_ads_*) was calculated from the ordinate of the origin. The Van’t Hoff plot is shown in Figure 12 and the *∆H°_ads_* and *∆S°_ads_* values are reported in Table 5.
(7)$$\ln K_{ads}=-\frac{\Delta H{{}^{\circ}}_{ads}}{RT}+\frac{\Delta S{{}^{\circ}}_{ads}}{R}+\ln\frac{1}{55.5}$$

The negative values of ∆*H°_ads_* for the H_2_SO_4_—CI system indicate not only that the adsorption process of Poly[VIMC4][Im], Poly[VIMC2][Br] and Poly[VIMC4][Br] had an exothermic nature, which is characteristic of physical adsorption [65], but also that the spontaneity of the adsorption process is limited by the temperature and that the protection of the metallic surface was favored at 298 K. On the other hand, the negative value of ∆*S°_ads_* reveals that the inhibiting molecules present in the acid-inhibitor electrolytic solution are adsorbed orderly on the surface [66].

The temperature effect on the corrosion inhibition of steel in the presence and absence of PILs can also be studied from the kinetic point of view by employing the Arrhenius equation shown in Equation (8). The activation energy (*E_a_*) is the energy that is necessary for a chemical reaction to be carried out and is related to the corrosion rate [67]:(8)$$\ln v_{corr}=-\frac{E_{a}}{RT}+\ln A$$where *v_corr_* represents the corrosion rate in g m^−2^ h^−1^ calculated with *i_corr_* values [68] and *A* is the Arrhenius pre-exponential factor. The kinetic parameters *E_a_* and *A* were obtained from the slope and ordinate of the origin in Figure 13, respectively.

Table 6 shows that *E_a_* augmented from 68.00 kJ/mol in the H_2_SO_4_ solution in the absence of CIs to 101.00 kJ/mol in the presence of PILs. According to the literature, if the *E_a_* values of the system in the presence of CIs are higher than those of the blank, the behavior can be associated with a physisorption process or also with the diminution of the inhibitor adsorption on the metallic surface as a consequence of the temperature increase; on the contrary, if the *E_a_* values are lower, the charge transfer from the inhibitor to the metallic surface to form coordinate covalent bonds, that is to say through a chemisorption process, is suggested [69].

An increase in the temperature of the acid medium implies the acceleration of the metal dissolution process and then higher *E_a_* values in the presence of CIs suggest that a higher energy physical barrier is formed by the electrostatic adsorption of Poly[VIMC4][Im], Poly[VIMC2][Br] and Poly[VIMC4][Br] molecules on the steel surface [59,70]. Such a physical barrier made of PILs limits the charge and mass transfer in the metallic interface [18]. On the other hand, *A* is related to the collision frequency between CI molecules and the acid medium and an increase in these values in the presence of PILs is associated with the increase in the system kinetic energy by effect of *T*.

Furthermore, although the *v_corr_* values diminished in the presence of PILs, the temperature increase reduced the adsorption process by the increase in the molecular kinetic energy of the corrosive medium, which provoked the growth of unprotected metallic surface fractions, as confirmed by the lowering of the *IE* values of the PILs from 298 to 328 K at constant *C_INH_*.

### 2.4. Surface Analysis

Figure 14 shows the surface analysis of the steel sample after its exposure to 1 M H_2_SO_4_ in the absence and presence of CIs at 308 K for 4 h by SEM. Figure 14a displays the blank micrograph with evident surface damage and heterogeneous morphology caused by *SO*_4_^2−^ and *H*_3_*O^+^* ions, which are characteristic of the corrosive medium and by the temperature effect. In the presence of PILs at 175 ppm (Figure 14b–d), a less damaged and more homogeneous morphology is observed; even some surface fractions spared by the corrosive attack can be seen, which reveals higher corrosion resistance originated by the adsorption of a CI protecting film; however, like in Figure 14a, there are also some sections that present localized corrosion promoted by the H_2_SO_4_ acid medium.

### 2.5. DFT Study

The quantum chemical calculation of the IL monomers helps better understand the inhibition mechanism by identifying the reactive sites of the CIs from the energy of the molecular orbitals and other quantum parameters. The optimized structures, molecular orbitals and molecular electrostatic potential (MEP) of the Poly[VIMC4][Im] and Poly[VIMC4][Br] monomers are shown in Table 7.

The CI behavior of the PIL monomers was studied by employing the energy of the highest occupied molecular orbital (*E_HOMO_*), the energy of the lowest unoccupied molecular orbital (*E_LUMO_*) and the charge distribution of the MEP (Table 7) [71,72].

In the case of Poly[VIMC4][Im], the species VBII presents HOMO in the imidazolate anion (site that cedes electrons) and LUMO in the imidazolium ring of the cation (site that accepts electrons). In addition, by being an IL species, it is confirmed that the negative charge is localized in the imidazolate anion and the positive charge in the imidazolium ring, thus identifying the molecule reactive sites. On the other hand, the VDAA monomer displayed the HOMO and LUMO orbitals distributed throughout its structure and because it is an organic species, the reactive sites are distinguished on the negative charges in C=O, positive charges in N–H_2_ and neutral charges in C–C. These results evidence that Poly[VIMC4][Im] has multiple adsorption sites that can increase the “anchoring” capacity of the molecule on the metallic surface. The results suggest that the imidazolate anions can work synergistically both with their imidazolium cations and diacetamide groups.

In the case of the VAA monomer of Poly[VIMC4][Br], the HOMO and LUMO distribution and the positive and negative charges are localized on the N and O heteroatoms, respectively, which indicates that the monomer presents donor-acceptor interactions; notwithstanding, some authors suggest that acrylamide participates mainly as an *e*^−^ donor with the metallic surface through π electrons present in the structure that promote the formation of coordination bonds with the *d* orbitals of Fe [71,73]. As for the VP monomer, it exhibits a distribution of the molecular orbitals and charge similar to that of VAA [74]. On the other hand, VBIB displays the HOMO and LUMO orbitals through the imidazolium ring as reported by Cui et al. [75], whereas the positive charge was localized in N and C=C of the imidazolium ring and the negative charge in the bromide anion. These results confirm that Br and C=C and the heteroatoms N and O work as reactive sites in the different polymeric blocks that favor the adsorption process through them.

The µ results indicate that both IL blocks (VBII > VBIB) present high values in comparison with the organic blocks (VAA > VP > VDAA), which reveals behavior associated with the ionic nature of the monomers that enhanced the adsorption process through dipole-dipole interactions between the CI and the metallic surface [71].

Table 8 reports the quantum parameters obtained from the monomers of the PILs. Based on the literature, the energy gap (Δ*G_L-H_*) is associated with the donor—acceptor activity of a molecule and is defined as the difference in LUMO and HOMO energies (Equation (9)):(9)$$\Delta G_{L-H}=E_{LUMO}-E_{HOMO}$$

Low Δ*G_L−H_* values suggest higher donor-acceptor activity of the monomers and based on the values shown in Figure 15, improved inhibiting behavior of the monomeric species of Poly[VIMC4][Im] can be considered.

## 3. Materials and Methods

Table 9 shows the chemical structures of the compounds evaluated as CIs: Poly(1-butyl-3-vinylimidazolium)imidazolate, Poly[VIMC4][Im], Poly(acrylamide-N-vinylpyrrolidone-1-ethyl-3-vinylimidazolium bromide), Poly[VIMC2][Br], and Poly(acrylamide-N-vinylpyrrolidone-1-butyl-3-vinylimidazolium bromide), Poly[VIMC4][Br]. The synthesis and characterization of the PILs was already reported in a previous work [28].

The corrosive acid medium employed in the present study was 1 M H_2_SO_4_, which was prepared by diluting reagent grade sulfuric acid (Sigma-Aldrich, Edo.de México, México) with deionized water. Afterward, PIL dilutions were prepared in the corrosive medium until obtaining concentrations of 100, 125, 150 and 175 ppm; the systems were evaluated at 308, 318 and 328 K.

API 5L X60 steel was used as the metallic sample whose chemical composition is displayed in Table 10. For the weight loss (WL) tests, the samples were abraded with 600 to 1200 grade SiC emery paper; then, the surfaces were cleaned with ethanol and deionized water, and finally, dried with nitrogen [76,77]. The initial mass of the samples was recorded before proceeding to their immersion in the acid-CI system. Gravimetric essays were carried out under static conditions without O_2_ extraction. The tests were performed at constant temperature (298, 308, 318 and 328 K) for 4 h employing a Carbolite LHT 4/60 oven. After the immersion process, the samples were extracted from the corrosive medium, washed, dried and weighed by following the ASTM G-01 standard [77].

The corrosion rate (*V_corr_*, mm year^−1^) and *IE*, obtained by means of the weight loss technique (*IE_WL_*, %), were calculated employing Equations (10) and (11) [76]:(10)$$V_{corr}=\frac{K\times \Delta W}{A\times t\times D}$$
(11)$$\%IE_{WL}=100\times \left(\frac{V_{corr}^{0}-V_{corr}^{CI}}{V_{corr}^{0}}\right)$$where *K* is a conversion factor equal to 8.76 × 10^4^, Δ*W* is the mass loss in g, *A* is the area in cm^2^, *t* is the time in h and *D* is the steel density equal to 7.86 gcm^−3^; the superindexes CI and 0 represent the presence and absence of CIs in the system, respectively [78].

As for the electrochemical evaluation, API 5L X60 steel was the working electrode (WE), which was mounted on Teflon tubes cured with epoxy resin, leaving a contact area of 0.2894 cm^2^. The open circuit potential (*E_OCP_*) of the WE was recorded for 1200 s at 308, 318 and 328 K in the absence and presence of CIs. The electrochemical measurements took place in a glass electrochemical cell with a three-electrode arrangement: platinum electrode (99.9%), Ag/AgCl in 3M KCl as the reference electrode and WE. The behavior of the PILs as CIs was analyzed electrochemically by means of an Autolab PGSTAT312N Potentiostat/Galvanostat controlled by the software NOVA 2.1.4, running linear polarization resistance (LPR) and potentiodynamic polarization (PDP) tests within an interval between ±25 mV and ±250 mV with respect to the *E_OCP_*, respectively, employing a scanning rate of 0.166 mVs^−1^.

Additionally, electrochemical impedance spectroscopy (EIS) tests were carried out at frequencies from 100 kHz to 100 mHz with a sinusoidal wave with an amplitude of 5 mV. In order to ensure the reproducibility of the electrochemical tests, all the experiments were run in triplicate [79,80,81]. The electrochemical parameters were obtained through linear regression of the LPR data and linear extrapolation of the PDP Tafel curves and EIS data by means of equivalent circuits. The IE of the PDP and EIS techniques were calculated employing Equations (12) and (13), respectively [82,83]:(12)$$\%IE_{PDP}=100\times \left[\frac{i_{corr}^{0}-i_{corr}^{CI}}{i_{corr}^{0}}\right]$$
(13)$$\%IE_{EIS}=100\times \left[\frac{Rp_{EIS}^{CI}-Rp_{EIS}^{0}}{Rp_{EIS}^{CI}}\right]$$where *i_corr_* is the current density in µAcm^−2^ and *Rp_EIS_* is the polarization resistance in Ωcm^2^ by the EIS technique.

In order to observe the surface morphology and identify the chemical elements present on the metallic surface in the absence and presence of PILs, the API 5L X60 steel samples were analyzed by scanning electron microscopy (SEM) employing a JEOL-JSM-6300 model of equipment. Before carrying out these analyses, the metallic samples were polished until reaching a mirror finishing and were submitted to immersion for 4 h in the absence and presence of 175 ppm of PILs at 308 K [76,77].

The inhibiting behavior exhibited by the PILs was supported by first principle energy calculations. The structures were optimized structurally, confirming the optimal position without symmetry restriction and in singlet state (M = 1). The computational calculations were developed under the density functional theory (DFT) through the software Gaussian 09W [84] based on the B3LYP/6-311++ theory level; Gauss view v6.0 was employed for visualization and input files. Once the optimal structure was produced, the molecular orbitals (MOs), molecular electrostatic potential (MEP) and dipolar moment (μ) of each PIL monomer were analyzed.

## 4. Conclusions

By performing the present research work, the temperature effect on the corrosion inhibition process of API 5L X60 steel in 1 M H_2_SO_4_ was confirmed by the presence of a protecting film consisting of the PILs Poly[VIMC4][Im], Poly[VIMC2][Br] and Poly[VIMC4][Br]. Based on the electrochemical and mass loss results, it was found that temperature is a variable that promotes the partial desorption process of CIs and, as a consequence, fewer fractions of protected metallic surface. Poly[VIMC4][Im] displayed a better adsorption process of the PIL derived from Poly[VIMC][Br] with a maximal *IE* of 75% at 175 ppm at 35 °C, which was attributed to the reticulated form of its chemical structure and to higher cationic participation of the vinylbutylimidazolium monomers. The evaluated compounds were classified as mixed-type CIs.

The SEM results supported the inhibition process of the PILs, revealing less surface damage through the adsorption and formation of an inhibiting film consisting of PILs, even with the temperature increase. The analysis of the molecular orbitals and molecular electrostatic potential of the monomers suggests that the PILs possess different reactive sites located mainly on the N and O heteroatoms present in their chemical structure.

## Figures and Tables

**Figure 1 ijms-24-06291-f001:**
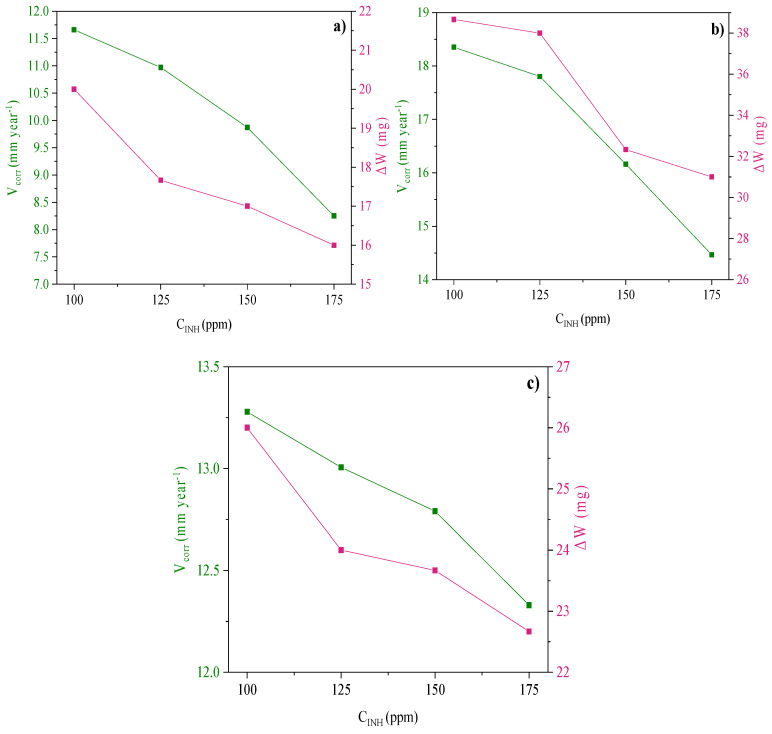
Corrosion rate (V_corr_) and mass loss (ΔW) of API 5L X60 steel after 4 h of immersion in 1 M H_2_SO_4_ and different *C_INH_*: (**a**) Poly[VIMC4][Im], (**b**) Poly[VIMC2][Br] and (**c**) Poly[VIMC4][Br] at 308 K.

**Figure 2 ijms-24-06291-f002:**
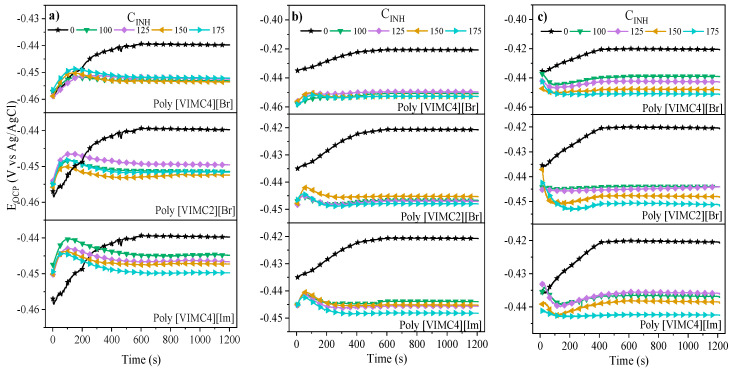
*E_OCP_* as a function of the immersion time of API 5L X60 steel in 1 M H_2_SO_4_ in the absence and presence of PILs as CIs at: (**a**) 308, (**b**) 318 and (**c**) 328 K.

**Figure 3 ijms-24-06291-f003:**
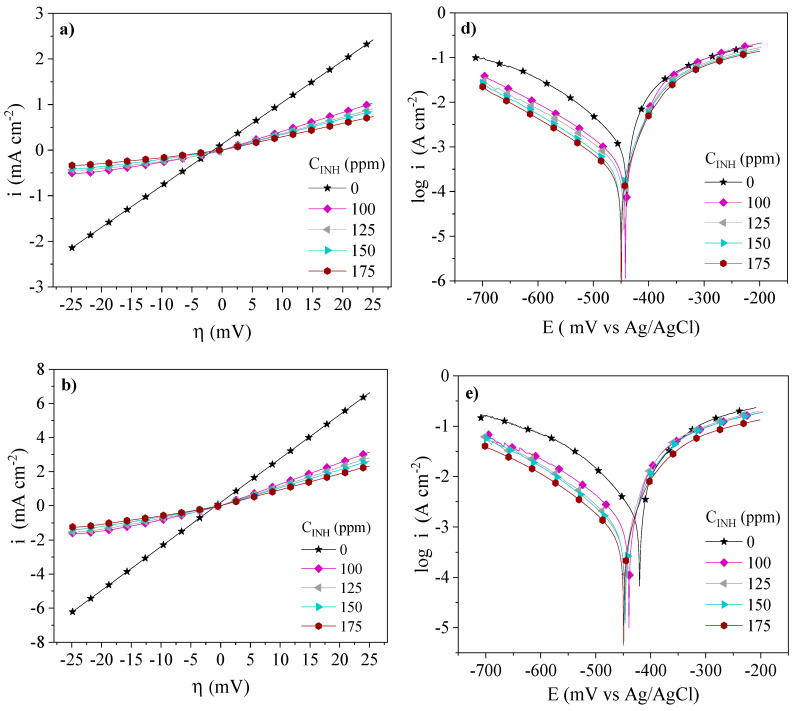
Electrochemical behavior of Poly[VIMC4][Im] as CI of API 5L X60 steel in 1 M H_2_SO_4_ by the LPR and PDP techniques: (**a**) LPR—308 K, (**b**) LPR—318 K, (**c**) LPR—328 K, (**d**) PDP—308 K, (**e**) PDP—318 K and (**f**) PDP—328 K.

**Figure 4 ijms-24-06291-f004:**
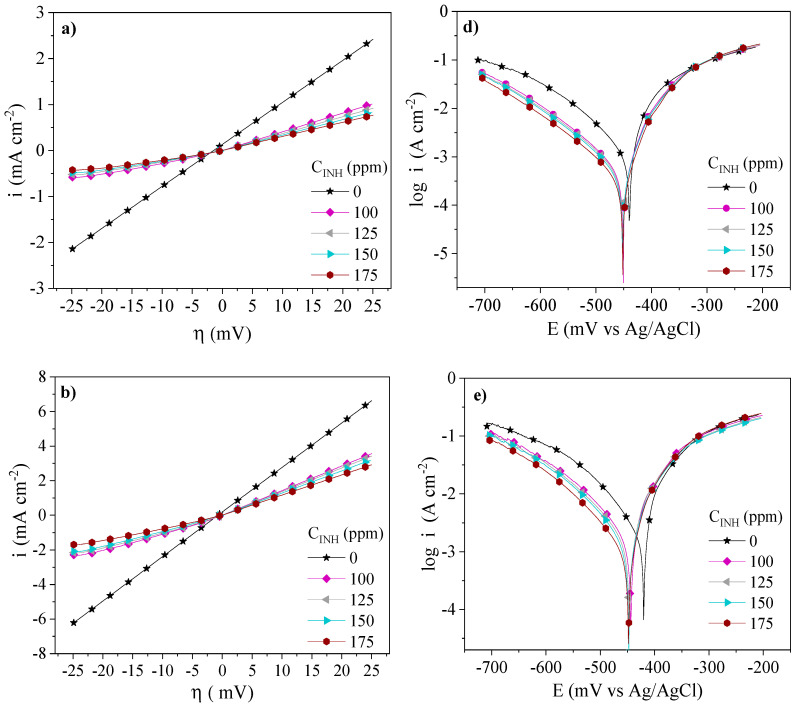
Electrochemical behavior of Poly[VIMC2][Br] as CI of API 5L X60 steel in 1 M H_2_SO_4_ by the LPR and PDP techniques: (**a**) LPR—308 K, (**b**) LPR—318 K, (**c**) LPR—328 K, (**d**) PDP—308 K, (**e**) PDP—318 K and (**f**) PDP—328 K.

**Figure 5 ijms-24-06291-f005:**
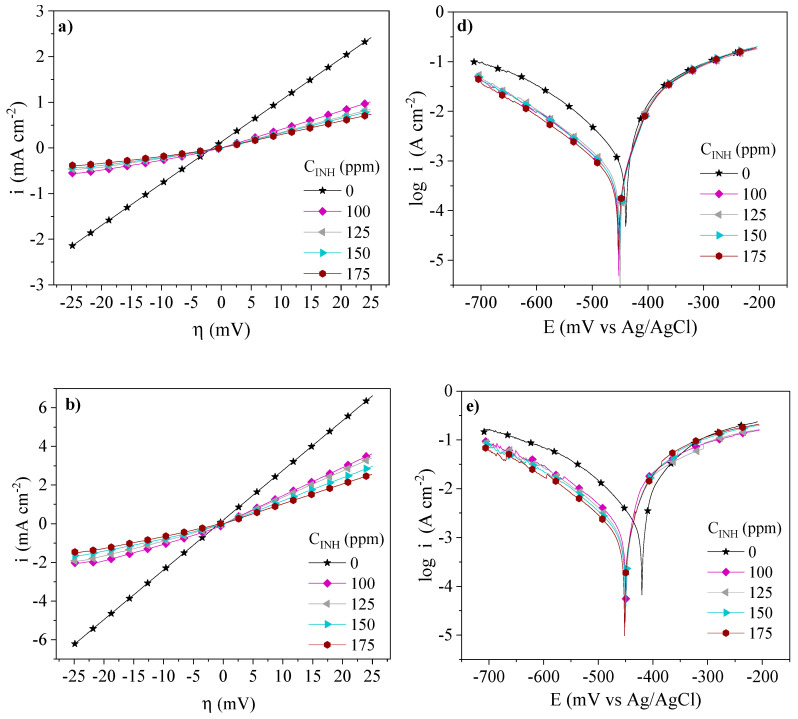
Electrochemical behavior of Poly[VIMC4][Br] as CI of API 5L X60 steel 1 M H_2_SO_4_ by the LPR and PDP techniques: (**a**) LPR—308 K, (**b**) LPR—318 K, (**c**) LPR—328 K, (**d**) PDP—308 K, (**e**) PDP—318 K, and (**f**) PDP—328 K.

**Figure 6 ijms-24-06291-f006:**
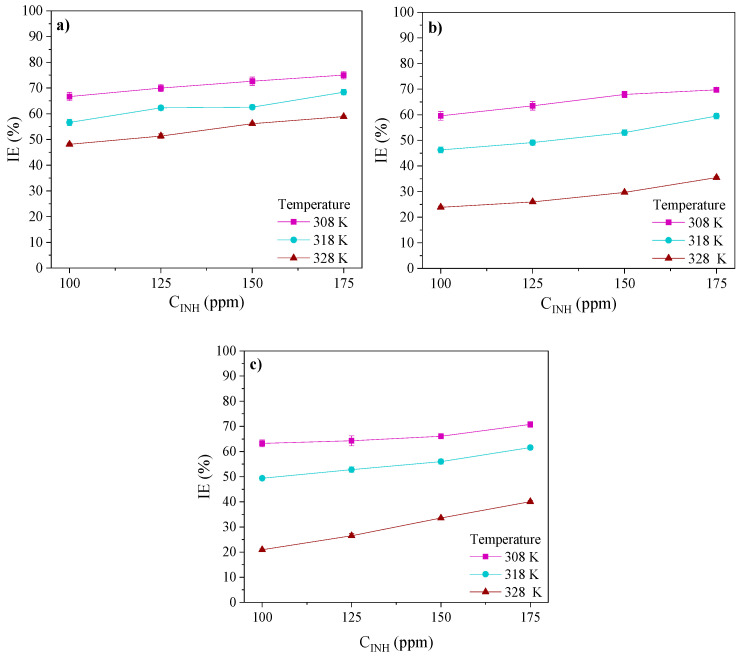
Inhibition efficiencies of API 5L X60 steel in 1 M H_2_SO_4_ in the absence and presence of PILs: (**a**) Poly[VIMC4][Im], (**b**) Poly[VIMC2][Br] and (**c**) Poly[VIMC4][Br].

**Figure 7 ijms-24-06291-f007:**
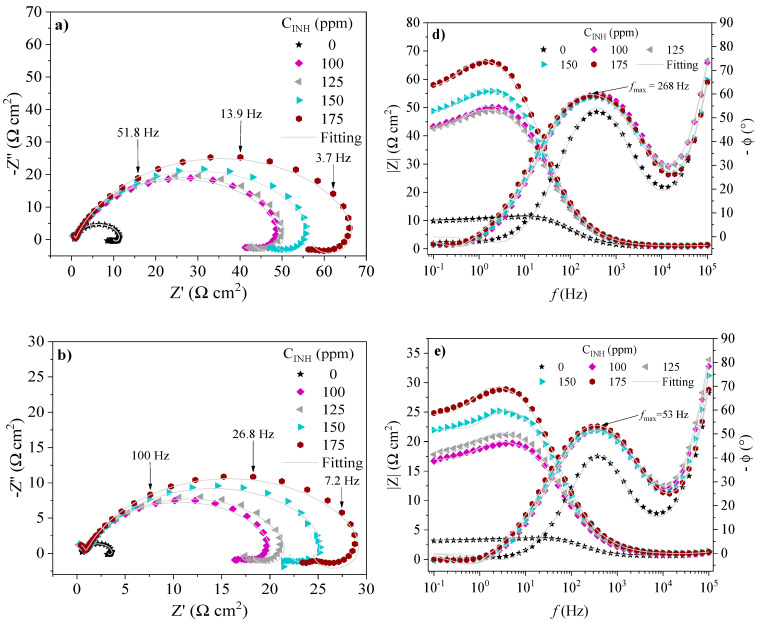
EIS spectra of API 5L X60 steel in 1 M H_2_SO_4_ in the presence of Poly[VIMC4][Im]: Nyquist [(**a**) 308 K, (**b**) 318 K and (**c**) 328 K] and Bode [(**d**) 308 K, (**e**) 318 K and (**f**) 328 K].

**Figure 8 ijms-24-06291-f008:**
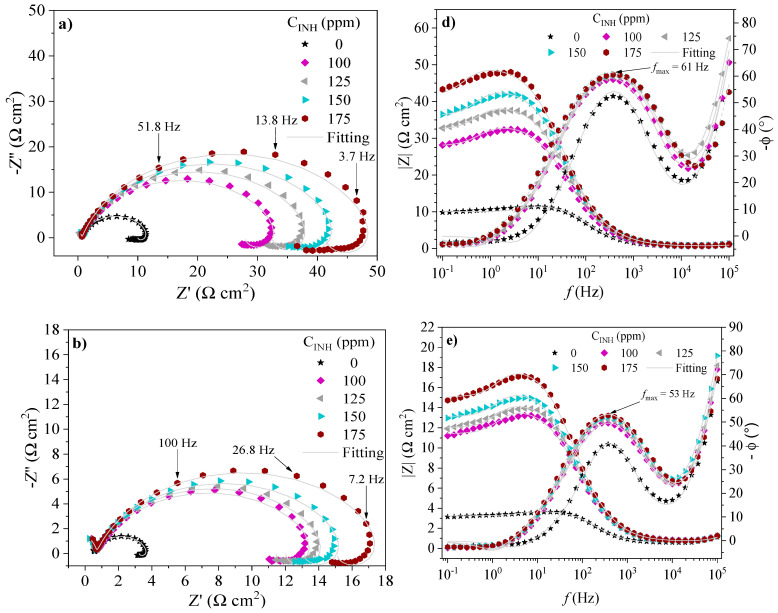
EIS spectra of API 5L X60 steel in 1 M H_2_SO_4_ in the presence of Poly[VIMC2][Br]: Nyquist [(**a**) 308 K, (**b**) 318 K and (**c**) 328 K] and Bode [(**d**) 308 K, (**e**) 318 K and (**f**) 328 K].

**Figure 9 ijms-24-06291-f009:**
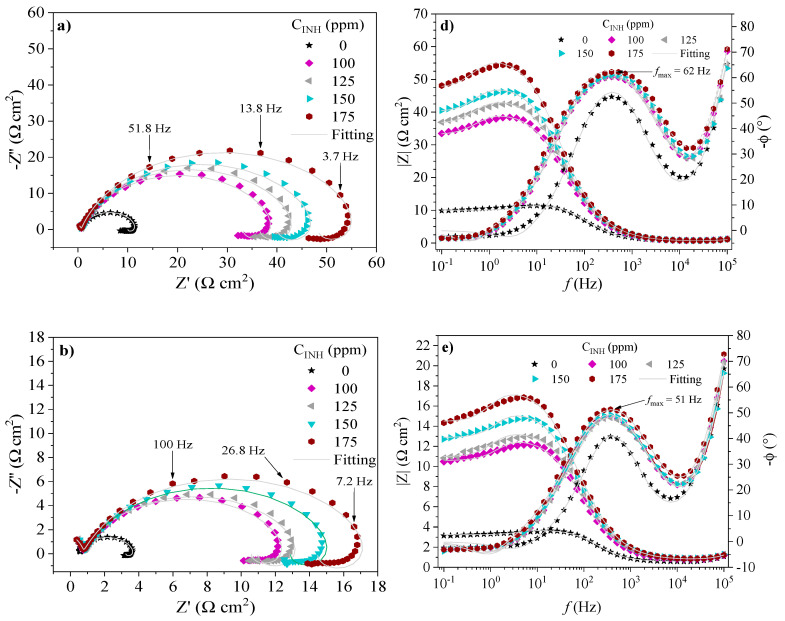
EIS spectra of API 5L X60 steel in 1 M H_2_SO_4_ in the presence of Poly[VIMC4][Br]: Nyquist [(**a**) 308 K, (**b**) 318 K and (**c**) 328 K] and Bode [(**d**) 308 K, (**e**) 318 K and (**f**) 328 K].

**Figure 10 ijms-24-06291-f010:**
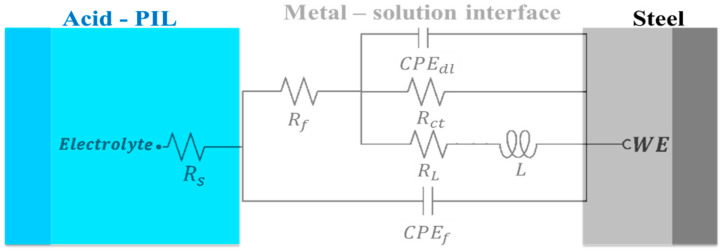
EEC for fitting the EIS experimental data of API 5L X60 steel in the acid—CI medium.

**Figure 11 ijms-24-06291-f011:**
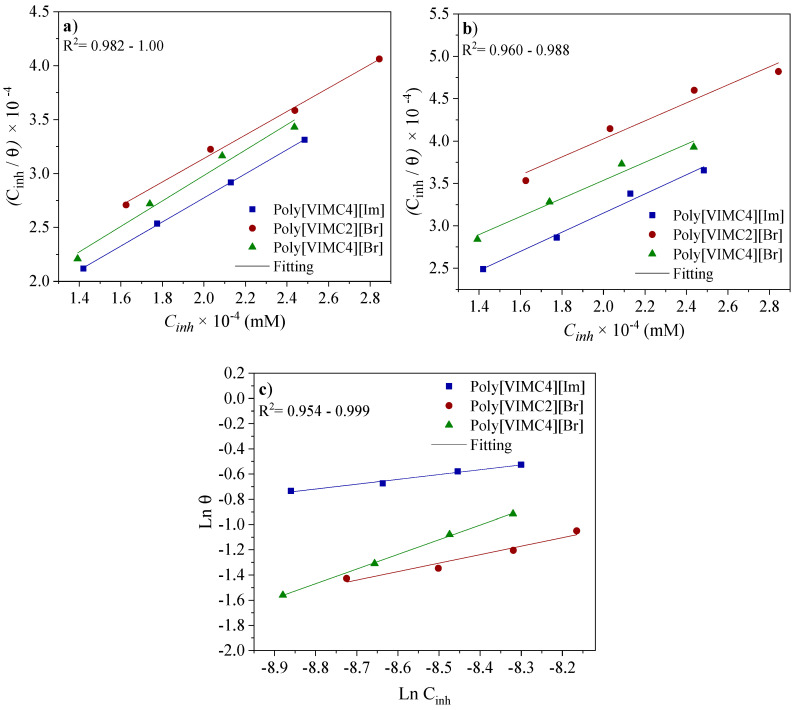
Adsorption isotherms of API 5L X60 steel in 1 M H_2_SO_4_ in the presence of PILs at different temperatures: (**a**) 308 K, (**b**) 318 K and (**c**) 328 K.

**Figure 12 ijms-24-06291-f012:**
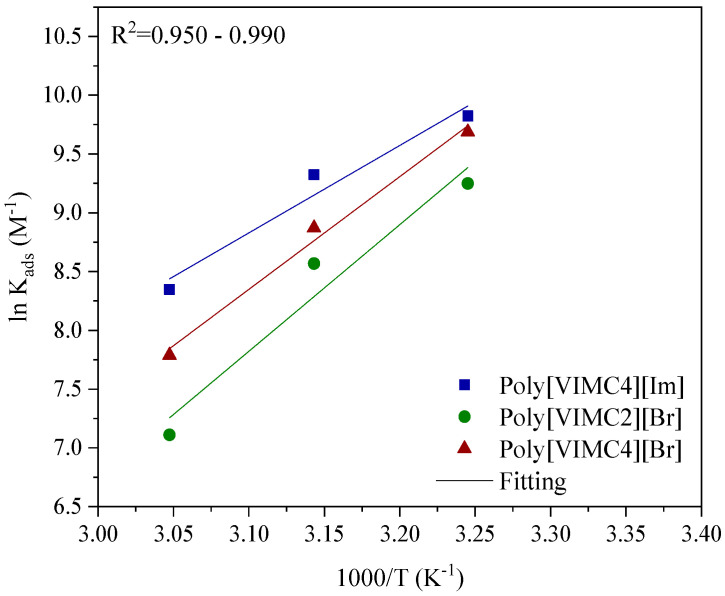
Van’t Hoff plot of API 5L X60 steel exposed to 1 M H_2_SO_4_—CIs.

**Figure 13 ijms-24-06291-f013:**
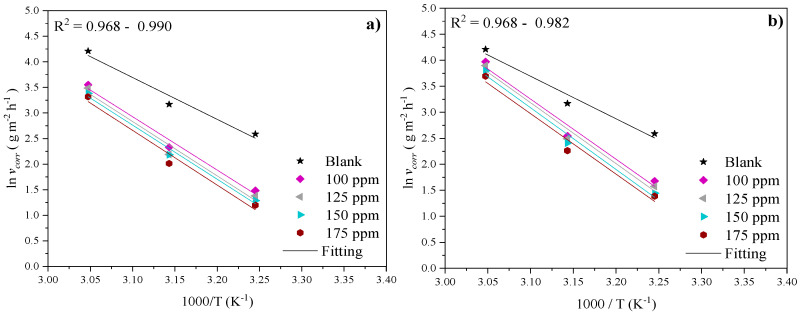
Arrhenius equation for API 5L X60 steel exposed to 1 M H_2_SO_4_ in the absence and presence of (**a**) Poly[VIMC4][Im], (**b**) Poly[VIMC2][Br] and (**c**) Poly[VIMC4][Br].

**Figure 14 ijms-24-06291-f014:**
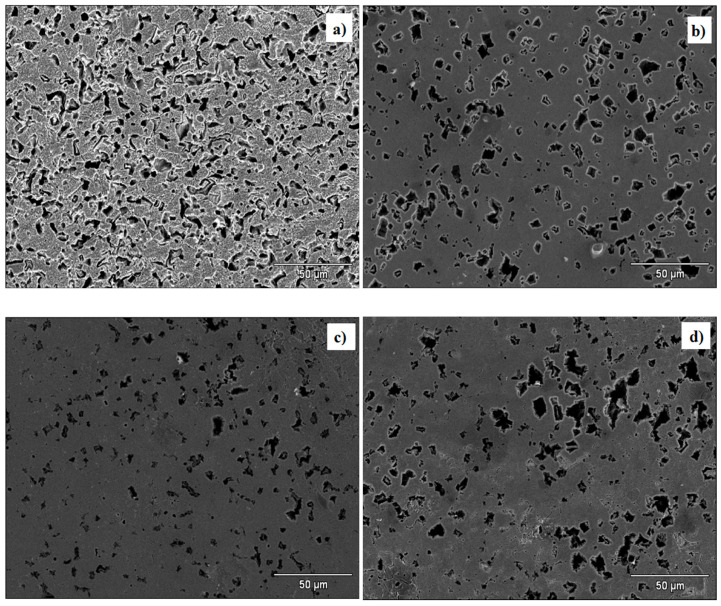
SEM micrographs of API 5L X60 steel exposed to 1 M H_2_SO_4_ at 308 K in the presence of: (**a**) 0 ppm de PIL, (**b**) 175 ppm of Poly[VIMC4][Im], (**c**) 175 ppm of Poly[VIMC2][Br] and (**d**) 175 ppm of Poly[VIMC4][Br].

**Figure 15 ijms-24-06291-f015:**
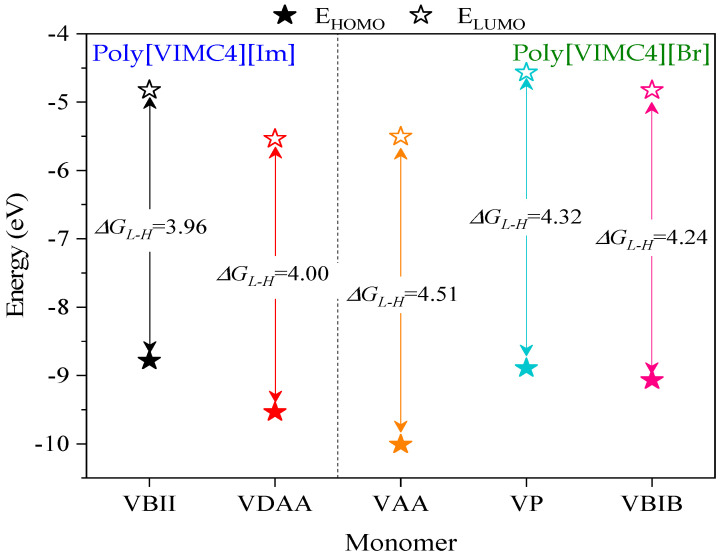
Representation of HOMO and LUMO energy levels and ΔG_L-H_ for the different monomers of their optimized geometries in water at B3LYP/6-311++.

**Table 1 ijms-24-06291-t001:** State of the art of polymers and PILs.

Polymers	Metal/Medium	*C* (ppm):*T* (K)/*IE* (%)	Ref.
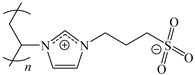	Carbon steel/0.5 M HCl	500 ppm:298 K/97.0%333 K/76.8%	[19]
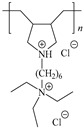	API X60 steel/15 wt.% HCl solution	1000 ppm:298 K/79%363 K/72%	[20]
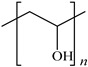	Carbon steel/1 M HCl	200 ppm:298 K/93%328 K/80%	[21]
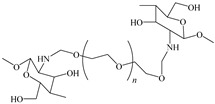	Mild steel/1 M sulfamic acid	250 ppm:308 K/90%338 K/59%	[22]
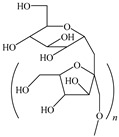	Mild steel/0.5 M H_2_SO_4_	1000 ppm:303 K/76.73%333 K/59.31%	[23]
6061Aluminumalloy/0.025 M HCl	1000 ppm:303 K/93.9%323 K/78.4%	[24]
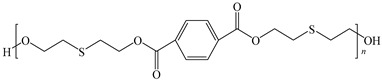	Carbon steel/0.1 M HCl	50 ppm:298 K/95.3%328 K/86.0%	[25]
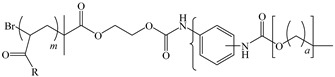	Mild steel/0.5 M H_2_SO_4_	1200 ppm:298 K/98%328 K/60%	[26]
1600 ppm:298 K/98.23%328 K/69.09%	[27]

**Table 2 ijms-24-06291-t002:** V_corr_ of API 5L X60 steel in 1 M H_2_SO_4_ with and without CIs by weight loss.

CI	*C_INH_*	*V_corr_* (mm Year^−1^)				*IE_WL_* (%)			
	(ppm)	298 K	308 K	318 K	328 K	298 K	308 K	318 K	328 K
Blank	0	14.08 ± 2.53	37.85 ± 3.20	103.51 ± 3.57	165.77 ± 5.24	–	–	–	–
Poly[VIMC4] [Im]	100	4.55 ± 0.20	11.66 ± 0.42	32.80 ± 3.05	79.71 ± 3.76	67.9 ± 5.9	69.2 ± 2.8	68.3 ± 3.1	51.9 ± 2.7
	125	4.50 ± 0.19	10.97 ± 0.55	29.71 ± 1.24	72.49 ± 4.75	68.2 ± 5.9	71.0 ± 2.9	71.3 ± 1.6	56.3 ± 3.2
	150	4.44 ± 0.26	9.87 ± 0.42	27.88 ± 0.80	59.83 ± 2.98	68.7 ± 5.9	73.9 ± 2.5	73.1 ± 1.2	63.9 ± 2.1
	175	3.31 ± 0.09	8.25 ± 0.67	26.67 ± 1.15	52.26 ± 1.54	76.6 ± 4.3	77.3 ± 2.6	74.2 ± 1.4	68.5 ± 1.4
Poly[VIMC2] [Br]	100	4.24 ± 0.21	18.35 ± 0.72	77.50 ± 3.36	128.25 ± 5.07	70.1 ± 5.6	51.5 ± 4.5	25.1 ± 4.1	22.6 ± 3.9
	125	3.77 ± 0.12	17.80 ± 1.15	64.42 ± 3.03	118.69 ± 4.80	73.4 ± 4.9	53.0 ± 5.0	37.8 ± 3.6	28.4 ± 3.7
	150	3.56 ± 0.26	16.16 ± 0.76	60.90 ± 3.10	110.55 ± 5.13	74.9 ± 4.9	60.2 ± 3.9	41.2 ± 3.6	33.3 ± 3.7
	175	3.24 ± 0.18	14.46 ± 0.66	56.79 ± 4.65	103.65 ± 1.04	77.1 ± 4.3	61.8 ± 3.7	45.1 ± 4.9	37.5 ± 2.1
Poly[VIMC4] [Br]	100	5.22 ± 0.19	13.28 ± 0.46	43.39 ± 1.65	122.25 ± 5.57	63.1 ± 6.8	64.9 ± 3.2	58.1 ± 2.2	26.3 ± 4.1
	125	2.97 ± 0.15	13.01 ± 0.45	39.17 ± 1.34	110.75 ± 2.98	79.1 ± 3.9	65.6 ± 3.1	62.2 ± 1.8	33.2 ± 2.8
	150	2.65 ± 0.22	12.79 ± 0.55	36.31 ± 1.39	91.80 ± 2.60	79.8 ± 4.0	66.2 ± 3.1	64.9 ± 1.8	44.6 ± 2.4
	175	2.53 ± 0.11	12.33 ± 0.62	32.55 ± 2.35	83.60 ± 4.72	81.1 ± 3.5	67.4 ± 3.0	68.6 ± 2.5	49.4 ± 3.3

**Table 3 ijms-24-06291-t003:** Electrochemical parameters of API 5L X60 steel in 1 M H_2_SO_4_—CIs at different temperatures by the LPR and PDP techniques.

*T*	*CI*	*C_INH_*	*Rp*	–*β_C_*	–*E_corr_*	*i_corr_*	*IE*
(K)		(ppm)	(Ω cm^2^)	(mV dec^−1^)	(mV)	(µA cm^−2^)	(%)
308	Blank	0	12.2 ± 0.4	123 ± 2	439 ± 2	1276 ± 26	–
	Poly [VIMC4][Im]	100	31.0 ± 1.2	128 ± 7	444 ± 3	425 ± 19	67 ± 2
		125	35.0 ± 0.8	127 ± 4	446 ± 3	383 ± 16	70 ± 1
		150	38.0 ± 1.5	128 ± 8	447 ± 3	349 ± 19	73 ± 2
		175	45.1 ± 1.5	133 ± 2	449 ± 1	319 ± 17	75 ± 1
	Poly [VIMC2][Br]	100	30.0 ± 0.9	113 ± 2	451 ± 0	516 ± 19	60 ± 2
		125	33.3 ± 0.7	108 ± 1	449 ± 2	466 ± 19	63 ± 2
		150	36.4 ± 0.9	113 ± 0	452 ± 2	409 ± 13	68 ± 1
		175	40.0 ± 0.5	115 ± 3	451 ± 1	386 ± 08	70 ± 1
	Poly [VIMC4][Br]	100	30.8 ± 0.5	111 ± 2	453 ± 1	469 ± 15	63 ± 1
		125	36.4 ± 0.6	113 ± 1	452 ± 1	456 ± 24	64 ± 2
		150	38.8 ± 1.0	115 ± 4	452 ± 1	433 ± 10	66 ± 1
		175	42.9 ± 1.6	111 ± 3	452 ± 1	373 ± 13	71 ± 1
318	Blank	0	3.8 ± 0.1	110 ± 2	420 ± 1	2282 ± 23	-
	Poly [VIMC4][Im]	100	10.0 ± 0.6	115 ± 6	442 ± 3	990 ± 26	57 ± 1
		125	11.0 ± 0.7	109 ± 8	442 ± 5	861 ± 19	62 ± 1
		150	12.0 ± 0.6	112 ± 6	442 ± 4	854 ± 19	63 ± 1
		175	13.5 ± 0.8	122 ± 6	448 ± 1	721 ± 23	68 ± 1
	Poly [VIMC2][Br]	100	8.2 ± 0.3	99 ± 5	445 ± 1	1226 ± 24	46 ± 1
		125	8.6 ± 0.7	104 ± 4	447 ± 1	1161 ± 20	49 ± 1
		150	9.2 ± 0.3	95 ± 4	445 ± 3	1072 ± 22	53 ± 1
		175	10.5 ± 0.5	101 ± 4	448 ± 0	925 ± 23	59 ± 1
	Poly [VIMC4][Br]	100	8.3 ± 0.2	100 ± 1	450 ± 1	1155 ± 14	49 ± 1
		125	9.1 ± 0.3	105 ± 11	449 ± 2	1077 ± 22	53 ± 1
		150	10.0 ± 0.3	107 ± 6	451 ± 1	1004 ± 21	56 ± 1
		175	12.0 ± 0.5	99 ± 4	452 ± 1	877 ± 17	62 ± 1
328	Blank	0	2.0 ± 0.1	139 ± 3	420 ± 3	6467 ± 29	-
	Poly [VIMC4][Im]	100	3.8 ± 0.4	129 ± 10	437 ± 2	3356 ± 30	48 ± 1
		125	4.4 ± 0.3	133 ± 6	436 ± 2	3147 ± 55	51 ± 1
		150	5.0 ± 0.2	132 ± 9	438 ± 1	2837 ± 23	56 ± 1
		175	5.6 ± 0.3	136 ± 7	445 ± 4	2659 ± 16	59 ± 1
	Poly [VIMC2][Br]	100	2.5 ± 0.2	99 ± 8	444 ± 3	4922 ± 21	24 ± 1
		125	2.8 ± 0.3	98 ± 3	443 ± 1	4788 ± 15	26 ± 1
		150	3.0 ± 0.1	118 ± 9	448 ± 0	4547 ± 12	30 ± 1
		175	3.3 ± 0.4	130 ± 4	451 ± 4	4173 ± 20	35 ± 1
	Poly [VIMC4][Br]	100	2.9 ± 0.1	119 ± 4	439 ± 2	5113 ± 10	21 ± 1
		125	3.1 ± 0.1	122 ± 10	442 ± 2	4752 ± 48	27 ± 1
		150	3.3 ± 0.1	127 ± 3	445 ± 3	4299 ± 11	34 ± 1
		175	3.5 ± 0.1	133 ± 3	451 ± 5	3876 ± 30	40 ± 1

**Table 4 ijms-24-06291-t004:** Electrochemical parameters of API 5L X60 steel in 1 M H_2_SO_4_—CIs at different temperatures by EIS.

*T*	*CI*	*C_INH_*	*χ^2^*	*R_s_*	*R_f_*	*C_f_*	*R_ct_*	*Y_0_*	*n*	*C_dl_*	*τ_dl_*	*R_L_*	*L*	*Rp_EIS_*	*IE_EIS_*
*K*		(ppm)		(Ω cm^2^)		(µF cm^−2^)	(Ω cm^2^)	(µS s^n^ cm^−2^)		(µF cm^−2^)	(ms)	(Ω cm^2^)	(H cm^2^)	(Ω cm^2^)	(%)
308	Blank	0	0.087	3.18 ± 0	3.90 ± 0.00	0.112 ± 0.001	12 ± 1	363 ± 7	0.88	177 ± 4	2.09 ± 0.05	55 ± 3	2 ± 1	16.8 ± 0.4	–
	Poly	100	0.018	3.18 ± 0	3.92 ± 0.01	0.126 ± 0.001	50 ± 0	346 ± 12	0.81	131 ± 4	6.60 ± 0.17	210 ± 4	79 ± 2	47.8 ± 0.3	64.8 ± 0.2
	[VIMC4][Im]	125	0.016	3.18 ± 0	3.99 ± 0.05	0.130 ± 0.002	52 ± 0	376 ± 15	0.80	138 ± 4	7.12 ± 0.17	213 ± 3	85 ± 3	48.7 ± 0.1	65.5 ± 0.1
		150	0.02	3.18 ± 0	4.13 ± 0.09	0.116 ± 0.002	56 ± 2	364 ± 27	0.80	133 ± 5	7.51 ± 0.03	231 ± 14	95 ± 9	52.6 ± 1.7	68.0 ± 1.0
		175	0.021	3.18 ± 0	4.14 ± 0.05	0.115 ± 0.000	69 ± 0	340 ± 8	0.80	129 ± 2	8.94 ± 0.12	292 ± 8	140 ± 2	63.3 ± 0.4	73.5 ± 0.2
	Poly	100	0.087	3.18 ± 0	3.90 ± 0.03	0.108 ± 0.002	36 ± 2	404 ± 8	0.81	151 ± 8	5.35 ± 0.04	152 ± 8	45 ± 1	35.8 ± 1.0	53.1 ± 1.9
	[VIMC2][Br]	125	0.025	3.18 ± 0	3.81 ± 0.02	0.123 ± 0.001	39 ± 0	385 ± 10	0.81	146 ± 3	5.64 ± 0.13	171 ± 7	59 ± 3	38.4 ± 0.5	56.2 ± 0.5
		150	0.031	3.18 ± 0	3.95 ± 0.11	0.117 ± 0.004	42 ± 2	370 ± 20	0.81	142 ± 8	5.96 ± 0.01	186 ± 6	63 ± 6	41.5 ± 1.5	59.4 ± 1.6
		175	0.100	3.18 ± 0	3.84 ± 0.01	0.083 ± 0.003	47 ± 2	377 ± 29	0.81	148 ± 9	6.99 ± 0.25	211 ± 21	112 ± 3	45.6 ± 1.9	63.1 ± 1.5
	Poly	100	0.035	3.18 ± 0	3.81 ± 0.02	0.116 ± 0.001	40 ± 1	359 ± 2	0.86	142 ± 5	5.67 ± 0.25	179 ± 5	61 ± 1	39.6 ± 0.4	57.6 ± 0.4
	[VIMC4][Br]	125	0.055	3.18 ± 0	3.85 ± 0.01	0.114 ± 0.006	43 ± 1	345 ± 8	0.84	136 ± 1	5.89 ± 0.10	191 ± 5	69 ± 3	41.9 ± 1.2	59.9 ± 1.1
		150	0.060	3.18 ± 0	3.82 ± 0.01	0.103 ± 0.002	46 ± 2	347 ± 12	0.81	138 ± 4	6.36 ± 0.21	209 ± 10	77 ± 2	44.8 ± 1.7	62.5 ± 1.5
		175	0.027	3.18 ± 0	3.81 ± 0.03	0.113 ± 0.001	55 ± 2	336 ± 13	0.81	135 ± 4	7.39 ± 0.13	273 ± 12	87 ± 6	52.6 ± 2.0	68.0 ± 1.2
318	Blank	0	0.044	3.18 ± 0	3.84 ± 0.08	0.106 ± 0.005	3 ± 0	696 ± 14	0.90	350 ± 7	1.13 ± 0.01	16 ± 1	1 ± 0	9.7 ± 0.1	-
	Poly [VIMC4][Im]	100	0.020	2.42 ± 0.05	3.18 ± 0.05	0.218 ± 0.007	21 ± 0	540 ± 15	0.79	165 ± 5	3.39 ± 0.05	82 ± 1	17 ± 2	22.0 ± 0.1	55.8 ± 0.3
		125	0.018	2.41 ± 0.03	3.17 ± 0.03	0.231 ± 0.007	22 ± 0	559 ± 8	0.78	167 ± 3	3.65 ± 0.03	85 ± 1	16 ± 0	23 ± 0.2	57.6 ± 0
		150	0.029	2.63 ± 0.01	3.55 ± 0.04	0.194 ± 0.003	25 ± 2	535 ± 8	0.78	161 ± 8	4.05 ± 0.09	99 ± 8	14 ± 3	26.3 ± 1.5	62.9 ± 2.1
		175	0.021	2.68 ± 0.06	3.61 ± 0.05	0.157 ± 0.005	29 ± 2	514 ± 29	0.78	156 ± 5	4.49 ± 0.11	107 ± 7	25 ± 3	29.0 ± 1.4	66.4 ± 1.6
	Poly [VIMC2][Br]	100	0.035	3.18 ± 0.00	3.87 ± 0.03	0.128 ± 0.001	14 ± 0	639 ± 6	0.80	202 ± 11	2.78 ± 0.08	56 ± 4	9 ± 2	18.1 ± 0.4	46.2 ± 1.1
		125	0.026	3.18 ± 0.00	3.87 ± 0.01	0.125 ± 0.004	14 ± 0	592 ± 28	0.81	198 ± 8	2.81 ± 0.10	59 ± 1	9 ± 0	18.5 ± 0.1	47.3 ± 0.3
		150	0.042	3.18 ± 0.00	3.81 ± 0.0	0.138 ± 0.002	16 ± 0	706 ± 11	0.79	216 ± 3	3.50 ± 0.05	59 ± 2	3 ± 0	19.7 ± 0.1	50.6 ± 0.2
		175	0.039	3.18 ± 0.00	3.82 ± 0.04	0.118 ± 0.003	19 ± 1	584 ± 8	0.81	185 ± 5	3.45 ± 0.09	90 ± 10	12 ± 1	22.5 ± 0.9	56.6 ± 1.7
	Poly [VIMC4][Br]	100	0.027	3.18 ± 0.00	3.91 ± 0.03	0.122 ± 0.00	13 ± 0	641 ± 13	0.81	206 ± 4	2.58 ± 0.03	51 ± 0	7 ± 0	17.1 ± 0.2	43.2 ± 0.8
		125	0.030	3.18 ± 0.00	3.95 ± 0.05	0.128 ± 0.003	13 ± 0	611 ± 11	0.81	189 ± 2	2.46 ± 0.04	53 ± 7	8 ± 0	18.0 ± 0.3	46.1 ± 0.9
		150	0.048	3.18 ± 0.00	3.95 ± 0.09	0.116 ± 0.002	15 ± 1	612 ± 33	0.81	197 ± 4	2.95 ± 0.18	56 ± 4	5 ± 0	19.0 ± 0.7	48.9 ± 1.8
		175	0.059	2.82 ± 0.37	3.55 ± 0.39	0.161 ± 0.040	21 ± 1	501 ± 10	0.80	162 ± 13	3.34 ± 0.18	116 ± 14	16 ± 5	23.9 ± 1.1	59.2 ± 1.8
328	Blank	0	0.016	2 ± 0.1	2.9 ± 0.1	0.3 ± 0.035	1.3 ± 0.1	961 ± 99	0.91	500 ± 18	0.66 ± 0.02	6.2 ± 0.4	0.14 ± 0.01	4.0 ± 0.1	-
	Poly [VIMC4][Im]	100	0.022	1.7 ± 0	2.8 ± 0.3	0.435 ± 0.067	6.2 ± 0.5	2029 ± 125	0.72	342 ± 21	2.11 ± 0.04	25 ± 2.4	1.25 ± 0.19	7.7 ± 0.1	47.8 ± 0.9
		125	0.029	1.4 ± 0.1	2.7 ± 0.1	0.454 ± 0.022	7 ± 0.2	2280 ± 116	0.7	349 ± 9	2.45 ± 0.10	27.4 ± 1.3	1.19 ± 0.27	8.3 ± 0.2	51.5 ± 1.3
		150	0.019	1.4 ± 0.1	3.1 ± 0.1	0.406 ± 0.025	7.2 ± 0.2	1976 ± 36	0.72	357 ± 5	2.59 ± 0.12	28.4 ± 1.6	1.68 ± 0.08	8.9 ± 0.1	54.5 ± 0.5
		175	0.031	1.4 ± 0.2	3.0 ± 0.1	0.477 ± 0.046	8.8 ± 0.4	2283 ± 170	0.70	384 ± 31	3.38 ± 0.10	32.7 ± 2.7	1.92 ± 0.39	9.9 ± 0.2	59.3 ± 1.0
	Poly [VIMC2][Br]	100	0.064	1.4 ± 0.2	2.3 ± 0.2	0.513 ± 0.08	3 ± 0.2	3220 ± 730	0.71	426 ± 12	1.29 ± 0.08	10.6 ± 0.7	0.42 ± 0.18	4.7 ± 0.2	14.2 ± 4.0
		125	0.050	1.5 ± 0	2.4 ± 0	0.504 ± 0.01	3.2 ± 0.1	2163 ± 68	0.75	373 ± 15	1.19 ± 0.09	12.5 ± 0.9	0.50 ± 0.06	4.9 ± 0.1	17.8 ± 1.4
		150	0.037	1.7 ± 0.1	2.6 ± 0	0.411 ± 0.011	3 ± 0.1	2040 ± 189	0.76	381 ± 24	1.14 ± 0.04	11 ± 0.4	0.46 ± 0.10	5.0 ± 0.1	19.2 ± 1.0
		175	0.025	2.1 ± 0.2	3.0 ± 0.1	0.292 ± 0.034	3.4 ± 0.1	2306 ± 69	0.74	385 ± 16	1.32 ± 0.08	12.3 ± 0.9	0.45 ± 0.07	5.7 ± 0.1	28.9 ± 1.6
	Poly [VIMC4][Br]	100	0.032	1.5 ± 0	2.4 ± 0	0.492 ± 0.009	3.8 ± 0.2	3969 ± 483	0.69	554 ± 37	1.38 ± 0.08	12.7 ± 1.1	0.52 ± 0.15	5.3 ± 0.2	23.3 ± 2.3
		125	0.035	1.8 ± 0.1	2.8 ± 0.1	0.366 ± 0.023	3.3 ± 0	4566 ± 425	0.67	526 ± 40	1.72 ± 0.11	14.3 ± 0.8	0.48 ± 0.13	5.5 ± 0.1	26.4 ± 1.2
		150	0.011	1.7 ± 0	3.1 ± 0.1	0.286 ± 0.011	3.1 ± 0.1	2772 ± 191	0.73	450 ± 15	2.08 ± 0.10	14.9 ± 0.6	0.85 ± 0.25	5.7 ± 0.1	29.0 ± 1.2
		175	0.031	1.6 ± 0	2.5 ± 0.1	0.533 ± 0.017	4.2 ± 0.1	2848 ± 511	0.72	455 ± 16	1.89 ± 0.12	17.3 ± 1.9	0.74 ± 0.16	5.9 ± 0.2	31.2 ± 2.2

**Table 5 ijms-24-06291-t005:** Thermodynamic parameters of API 5L X60 steel in the presence of 1 M H_2_SO_4_—CIs at different temperatures.

CI	*T* (K)	*R^2^*	Slope	*K_ads_* (L mol^−1^)	*−*Δ*G°_ads_* (kJ mol^−1^)	*−*Δ*H°_ads_* (kJ/mol)	*−*Δ*S°_ads_* (J/mol K)
Poly[VIMC4][Im]	308	1.000	1.117	1.85 × 10^4^	35.46	61.81	84.82
	318	0.988	1.129	1.12 × 10^4^	35.29		
	328	0.985	0.382	4.22 × 10^3^	30.64		
Poly[VIMC2][Br]	308	0.996	1.089	1.04 × 10^4^	33.98	89.48	178.97
	318	0.960	1.062	5.26 × 10^3^	33.29		
	328	0.954	0.673	8.25 × 10^2^	24.37		
Poly[VIMC4][Br]	308	0.982	1.181	1.61 × 10^4^	35.11	79.69	144.24
	318	0.975	1.066	7.12 × 10^3^	34.09		
	328	0.999	1.162	1.91 × 10^3^	28.69		

**Table 6 ijms-24-06291-t006:** Kinetic parameters of the inhibition process of API 5L X60 steel in 1 M H_2_SO_4_ with and without CIs.

	Poly[VIMC4][Im]		Poly[VIMC2][Br]		Poly[VIMC4][Br]	
*C_INH_* (ppm)	*E_a_* (kJ mol^−1^)	*A* (g m^−2^ h^−1^)	*E_a_* (kJ mol^−1^)	*A* (g m^−2^ h^−1^)	*E_a_* (kJ mol^−1^)	*A* (g m^−2^ h^−1^)
0	68.00	4.12 × 10^12^	68.00	4.12 × 10^12^	68.00	4.12 × 10^12^
100	86.67	2.03 × 10^15^	95.13	6.49 × 10^16^	100.13	4.11 × 10^17^
125	88.29	3.37 × 10^15^	97.59	1.54 × 10^17^	98.22	1.88 × 10^17^
150	87.92	2.73 × 10^15^	101.00	5.12 × 10^17^	96.19	8.12 × 10^16^
175	88.89	3.54 × 10^15^	99.76	2.91 × 10^17^	98.09	1.47 × 10^17^

**Table 7 ijms-24-06291-t007:** Optimized structures, HOMO, LUMO and molecular electrostatic potential mapping of monomers obtained at the B3LYP/6-311++ theory level in aqueous medium.

PIL	Monomer	Optimized Structure	HOMO	LUMO	MEP
Poly[VIMC4][Im]	Vinyl-butylimidazolium imidazolate (VBII)	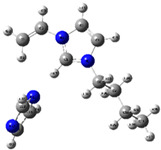	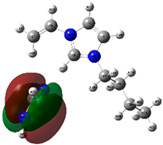	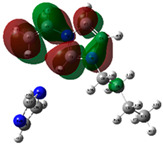	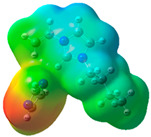
	Vinyl-diacetamide (VDAA)	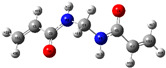	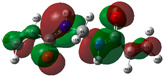	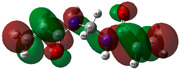	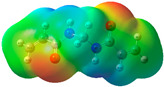
Poly[VIMC4][Br]	Vinylacrylamide (VAA)	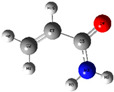	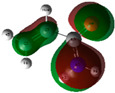	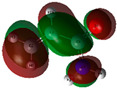	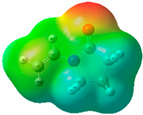
	Vinylpyrrolidone (VP)	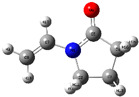	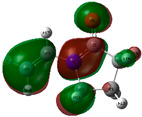	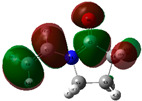	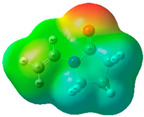
	Vinyl-butylimidazolium bromide (VBIB)	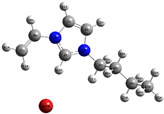	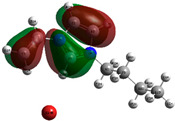	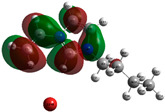	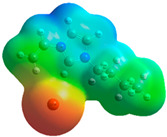

**Table 8 ijms-24-06291-t008:** Quantum parameters obtained from PIL monomers at B3LYP/6-311++ level in aqueous medium.

PIL	Monomers	–*E_HOMO_* (eV)	–*E_LUMO_* (eV)	*µ* (debye)
Poly[VIMC4][Im]	VBII	8.78	4.82	20.82
	VDAA	9.54	5.54	1.03
Poly[VIMC4][Br]	VAA	10.01	5.50	6.14
	VP	8.89	4.57	5.94
	VBIB	9.07	4.83	19.51

**Table 9 ijms-24-06291-t009:** Chemical structures of the PILs evaluated as CIs.

Abbreviation	Name	Chemical Structure
Poly[VIMC4][Im]	Poly(1-butyl-3-vinylimidazolium)imidazolate	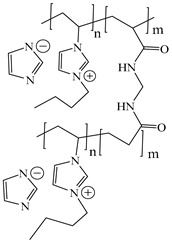
Poly[VIMC2][Br]	Poly(acrylamide-*N*-vinylpyrrolidone-1-ethyl-3-vinylimidazolium bromide)	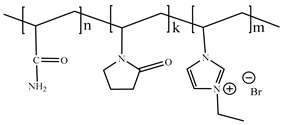
Poly[VIMC4][Br]	Poly(acrylamide-*N*-vinylpyrrolidone-1-butyl-3-vinylimidazolium bromide)	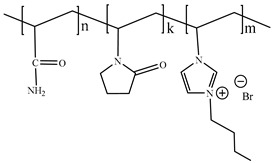

**Table 10 ijms-24-06291-t010:** Percentage chemical composition of API 5L X60 steel [78].

C	Mn	Si	Cr	Mo	V	Cu	Ni	Al	P	S	Ti	Nb	Fe
0.14	1.04	0.25	0.07	0.08	0.03	0.03	0.05	0.026	0.014	0.011	0.015	0.001	balance

## Data Availability

The data presented in this study are available on request from the corresponding author.

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
