# Peer review of "Temperature Effect on the Corrosion Inhibition of Carbon Steel by Polymeric Ionic Liquids in Acid Medium"

_ijms, 2023, doi:10.3390/ijms24076291_

Round 1

Reviewer 1 Report

Comments and suggestions:

1. Abstract- “All the tested PILs displayed mixed-type corrosion inhibitor (CI) behavior” – Mention the results here.

2. “Keywords: Temperature; Corrosion inhibition; Poly(Ionic Liquid); EIS; Polarization; SEM; DFT. "- Use different keywords, which are not used in title.

3. “…., industry sectors 33 such as exploration and production of oil and gas, oil refineries, production of chemical products, heavy industry, water treatment and product additive industry are normally benefited by the protective action of CIs” --- I suggest the author, to discuss a paragraph related to water treatment techniques. The authors are recommended to check the below related references, which will improve the supporting information.

Journal of Molecular Liquids 317, 2020, 113916

Polymers 14 (5), 2022, 914

Journal of Hazardous Materials 365, 2019, 759-770

Environmental research 2019, 170, 389-397

4. “In the present research work, the temperature effect on the corrosion inhibition process of 15 API 5L X60 steel in 1 M H2SO4 by employing three vinylimidazolium poly(ionic liquid)s (PILs) was 16 studied by means of electrochemical techniques, surface analysis and computational simulation.” – The authors need to discuss about the applications of various electrochemical techniques in introduction section and include the work mention in below references.

Journal of Energy Storage 60, 2023, 106554

Sensors and Actuators B: Chemical 146 (1), 2010, 314-320

Fuel 330, 2022, 125530

Colloids and Surfaces A: Physicochemical and Engineering Aspects 647, 2022, 129077

Journal of Electroanalytical Chemistry 918, 2022, 116521

5. “The diminution of the IEs with the T increase implies higher kinetic energy in the redox reactions and, as a consequence, the increase in the steel anodic dissolution, which provoked the desorption of the PIL molecules and less covered fraction” – Explain the reason in brief.

6. “shows the adsorption process of the PILs Poly[VIMC4][Im], Poly[VIMC2][Br] and Poly[VIMC4][Br] on the steel surface at 308 and 318 K, which can be described by the Langmuir adsorption isotherm (Equation 4) due to the correlation coefficients (R2 ) close to unity” – Need supporting information,  

Journal of Cleaner Production 244, 2020, 118867

Biomass Conv. Bioref. (2023). https://doi.org/10.1007/s13399-022-03711-7

7. “The corrosive acid medium employed in the present study was 1 M H2SO4, which was prepared by diluting reagent grade sulfuric acid with deionized water..”- Provide the details of the supplier including city and country and also do same for all chemicals used.

8. The English quality not up to the mark. All the typos and grammar need to check thoroughly in the manuscript.

Author Response

Answers to the Reviewers’ questions

The authors thank the Reviewers for their precious time devoted to review the manuscript ijms-2270086. All the insightful observations and suggestions were taken into account to enhance the submitted paper according to the quality standards set by the prestigious International Journal of Molecular Sciences.

Reviewer 1

Comments and suggestions

  1. Abstract- “All the tested PILs displayed mixed-type corrosion inhibitor (CI) behavior” – Mention the results here.

Answer: Thank you very much for the suggestion. It was taken into account accordingly.

  1. Keywords: Temperature; Corrosion inhibition; Poly(Ionic Liquid); EIS; Polarization; SEM; DFT. "- Use different keywords, which are not used in title.

Answer: We do thank the Reviewer’s suggestion. The corresponding changes were performed in the manuscript.

  1. “…., industry sectors 33 such as exploration and production of oil and gas, oil refineries, production of chemical products, heavy industry, water treatment and product additive industry are normally benefited by the protective action of CIs” --- I suggest the author, to discuss a paragraph related to water treatment techniques. The authors are recommended to check the below related references, which will improve the supporting information.

Journal of Molecular Liquids 317, 2020, 113916

Polymers 14 (5), 2022, 914

Journal of Hazardous Materials 365, 2019, 759-770

Environmental research 2019, 170, 389-397

Answer: Thank you very much for the suggestion. We carried out the research of the articles mentioned by the Reviewer and by reading them, came to the conclusion that the reported topics are not related to the study and use of corrosion inhibitors to mitigate the damage of steel in corrosive media. Although these works are very interesting in the water treatment area, they do not deal with the use of ionic liquids. In addition to not having any connection with our research work, we found that these titles feature common authors as shown below: 

  • Oxygenated functionalities enriched MWCNTs decorated with silica coated spinel ferrite – A nanocomposite for potentially rapid and efficient de-colorization of aquatic environment. Zeid Abdullah Alothman, Saikh Mohammad Wabaidur, Moonis Ali Khan, Masoom Raza Siddiqui, Marta Otero,  Byong-Hun Jeon, Afnan Ali Hussain Hakami, Journal of Molecular Liquids 317, 2020, 113916
  • Mohammad Azam, Saikh Mohammad Wabaidur, Mohammad Rizwan Khan, Saud I. Al-Resayes and Mohammad Shahidul Islam. Heavy Metal Ions Removal from Aqueous Solutions by Treated Ajwa Date Pits: Kinetic, Isotherm, and Thermodynamic Approach. Polymers 14 (5), 2022, 914
  • Moonis Ali Khan, Marta Otero, Mohsin Kazi, Ayoub Abdullah Alqadami, Saikh Mohammad Wabaidur, Masoom Raza Siddiqui, Zeid Abdullah Alothman, Sadia Sumbul, Unary and binary adsorption studies of lead and malachite green onto a nanomagnetic copper ferrite/drumstick pod biomass composite, Journal of Hazardous Materials, Volume 365, 2019, Pages 759-770,
  • Imran Ali, Omar M.L. Alharbi, Zeid A. ALOthman, Amal Mohammed Al-Mohaimeed, Abdulrahman Alwarthan, Modeling of fenuron pesticide adsorption on CNTs for mechanistic insight and removal in water, Environmental Research, Volume 170, 2019, Pages 389-397,

  1. “In the present research work, the temperature effect on the corrosion inhibition process of 15 API 5L X60 steel in 1 M H2SO4 by employing three vinylimidazolium poly(ionic liquid)s (PILs) was 16 studied by means of electrochemical techniques, surface analysis and computational simulation.” – The authors need to discuss about the applications of various electrochemical techniques in introduction section and include the work mention in below references.

Journal of Energy Storage 60, 2023, 106554

Sensors and Actuators B: Chemical 146 (1), 2010, 314-320

Fuel 330, 2022, 125530 

Colloids and Surfaces A: Physicochemical and Engineering Aspects 647, 2022, 129077

Journal of Electroanalytical Chemistry 918, 2022, 116521

Answer: As it is well known, the study of organic and inorganic molecules and ionic liquids as corrosion inhibitors is carried out by means of electrochemical techniques, where the most employed are polarization resistance, Tafel polarization and electrochemical impedance. This is due to their contribution to the understanding of the kinetics of the redox processes that take place during the corrosion phenomenon. As for the articles cited in the Introduction, all of them feature the previously mentioned techniques and support the subject matter that is the use of ionic liquids as corrosion inhibitors. Once again, relevant and high quality articles are recommended to be cited in our work, but we consider that they do not have anything to do with the study of ionic liquids; in addition, by checking the suggested references, we found similarities with the previously proposed works as follows:

  • Ragupathy Dhanusuraman, Priyanka Chahal, Asha Raveendran, Maimur Hossain, Razan A. Alshgari, Saikh Mohammad Wabaidur, Muthusankar Eswaran. Facile fabrication of platinum loaded poly(2,5-dimethoxy aniline)/activated carbon ternary nanocomposite as an efficient electrode material for high performance supercapacitors, Journal of Energy Storage, Volume 60, 2023, 106554,
  • Zeid Abdullah Alothman, Nausheen Bukhari, Saikh Mohammad Wabaidur, Sajjad Haider, Simultaneous electrochemical determination of dopamine and acetaminophen using multiwall carbon nanotubes modified glassy carbon electrode, Sensors and Actuators B: Chemical, Volume 146, Issue 1, 2010, Pages 314-320,
  • Vishnu Sankar Sivasankarapillai, N. Veni Keertheeswari, Priyanka Chahal, Saikh Mohammad Wabaidur, Vinoth Kumar Ponnusamy, Ragupathy Dhanusuraman, Facile electrodeposition fabrication of raspberry-like gold microspheres decorated polydiphenylamine nanohybrid coated electrode for efficient direct methanol fuel cell application, Fuel, Volume 330, 2022, 125530.
  • Jaysiva Ganesamurthi, Ragurethinam Shanmugam, Tse-Wei Chen, Shen-Ming Chen, Muthukutty Balamurugan, Zhe-Wei Gan, Masoom Raza Siddiqui, Saikh Mohammad Wabaidur, Mohammad Ajmal Ali, NiO/ZnO binary metal oxide based electrochemical sensor for the evaluation of hazardous flavonoid in biological and vegetable samples, Colloids and Surfaces A: Physicochemical and Engineering Aspects, Volume 647, 2022, 129077.
  • Subramaniyan Pulikkutty, Natesan Manjula, Tse-Wei Chen, Shen-Ming Chen, Bih-Show Lou, Masoom Raza Siddiqui, Saikh Mohammad Wabaidur, Mohammad Ajmal Ali, Fabrication of gadolinium zinc oxide anchored with functionalized-SWCNT planted on glassy carbon electrode: Potential detection of psychotropic drug (phenothiazine) in biotic sample, Journal of Electroanalytical Chemistry, Volume 918, 2022, 116521.

  1. “The diminution of the IEs with the T increase implies higher kinetic energy in the redox reactions and, as a consequence, the increase in the steel anodic dissolution, which provoked the desorption of the PIL molecules and less covered fraction” – Explain the reason in brief.

Answer: It is known that the molecular diffusion of ions toward a metallic substrate increases with the temperature. In this case, the diffusion of hydronium and sulfate ions toward the steel surface, promotes an increase in the kinetics of the electrochemical reactions; such mechanism provokes, in turn, that the protecting film becomes weak due to the partial desorption of the PIL macromolecules from the steel surface, which means that there will be less surface fractions covered by PILs. In this way, metal surface active sites are available and vulnerable to be attacked by medium corrosive ions, thus increasing the rate of the redox reactions.  

  1. “shows the adsorption process of the PILs Poly[VIMC4][Im], Poly[VIMC2][Br] and Poly[VIMC4][Br] on the steel surface at 308 and 318 K, which can be described by the Langmuir adsorption isotherm (Equation 4) due to the correlation coefficients (R2 ) close to unity” – Need supporting information, 

Answer: The information was supported as suggested by the Reviewer.

  1. “The corrosive acid medium employed in the present study was 1 M H2SO4, which was prepared by diluting reagent grade sulfuric acid with deionized water..”- Provide the details of the supplier including city and country and also do same for all chemicals used.

Answer: The observation was taken into account as suggested by the Reviewer.

Reviewer 2 Report

In this manuscript, the authors studied the effect of three vinylimidazolium poly(ionic liquid)s corrosion inhibitors for API 5L X60 steel in 1 M H2SO4. The corrosion inhibition of the tested inhibitors was analyzed by different electrochemical methods at different temperatures. A computational chemical approach was used to support the experimental results. The topic is interesting from a corrosion point of view. The experimental campaign described in the paper is rich and interesting, and the experimental outcomes are properly discussed.

My overall suggestion is to accept the paper after some improvements:

1.      Line118. Vcor must be written what it means. 2.      Report the corrosion inhibition efficiency in Tables 2 and 4 and explain the results. 3.      It is better to show the EIS results in the frequency from 10kHz to 0.1 Hz. At higher frequencies and temperatures, in the Nyquist plots, you have an extra arc. This is likely due to your reference electrode. 4.      Plus, why did the solution resistance change with the Cinh at higher temperatures? 5.      Equation 3. If you reported fmax, so you have also to display the fmax on the Nyquist plots. 6.      Line278. “Rs did not display any significant change, which revealed that the corrosive systems underwent minimal ohmic drop.” a.       Actually, from the graphs, it seems the opposite, especially at high temperatures 7.      Caption figure 14. Write the temperature also in here. 8.      SEM analysis. Why did you show only the 308K temperature results? 9.      Line 511. Ag/AgCl reference electrode. Write the concentration of KCl. 10.  No surface morphology results, except SEM, were presented. Product analysis should be conducted, including XPS, etc

Author Response

Answers to the Reviewers’ questions

The authors thank the Reviewers for their precious time devoted to review the manuscript ijms-2270086. All the insightful observations and suggestions were taken into account to enhance the submitted paper according to the quality standards set by the prestigious International Journal of Molecular Sciences.

Revisor 2:

  1. Vcor must be written what it means

Answer: The observation was taken into account.

  1. Report the corrosion inhibition efficiency in Tables 2 and 4 and explain the results.

Answer: Thank you very much for the suggestion and it was taken into account accordingly.

  1. It is better to show the EIS results in the frequency from 10kHz to 0.1 Hz. At higher frequencies and temperatures, in the Nyquist plots, you have an extra arc. This is likely due to your reference electrode. 

Answer: We carried out the reduction of the frequency interval as suggested by the Reviewer, however, as it can be observed in the figure below, at 328 K, the high frequency loop is still evident. Furthermore, if the scale adjustment is performed as suggested, the spectrum shows less experimental data. We consider that the effect of the reference electrode does not affect the EIS spectra, because, at other temperatures, there is no important contribution to the extra loop at high frequencies.

  1. Plus, why did the solution resistance change with the Cinh at higher temperatures? 

Answer: As observed in Table 3, in the manuscript, the variation of the Rs values is minimal (at 308 K, on average, there is a value of 3.18 Ω cm2, at 318 K of 2.935 Ω cm2 and at 328 K of 1.63 Ω cm2); for this reason, we consider that the observation regarding the extra loop at high frequencies revealed in the Nyquist plots is related to the Rf and CPEf components attributed to a film formed on the metallic surface (where corrosion products can be found) with different dielectric properties; this fact also confirms that at high temperatures, there are higher corrosion rates. 

  1. Equation 3. If you reported fmax, so you have also to display the fmax on the Nyquist plots. 

Answer: We added the requested information to the impedance spectra as suggested by the Reviewer.

  1. Line278. “Rs did not display any significant change, which revealed that the corrosive systems underwent minimal ohmic drop.” a.       Actually, from the graphs, it seems the opposite, especially at high temperatures.

Answer: This observation was answered in question 4.

  1. Caption figure 14. Write the temperature also in here.

Answer: The observation was taken into account as follows: Figure 14. SEM micrographs of API 5L X60 steel exposed to 1 M H2SO4 at 308 K in the presence of: (a) 0 ppm of PIL, (b) 175 ppm of Poly[VIMC4][Im], (c) 175 ppm of Poly[VIMC2][Br] and (d) 175 ppm of Poly[VIMC4][Br].

  1. SEM analysis. Why did you show only the 308K temperature results? 

Answer: The SEM analysis was carried out at 308 K, because the maximal inhibition efficiency values were obtained at this temperature: 75%  for  Poly  [VIMC4][Im],  70%  for  Poly  [VIMC2][Br]  and  71  %  for Poly[VIMC4][Br].

  1. Line 511. Ag/AgCl reference electrode. Write the concentration of KCl. 

Answer: The concentration for the Ag/AgCl reference electrode was 3M KCl and it is reported in the corrected version of the manuscript.

  1. No surface morphology results, except SEM, were presented. Product analysis should be conducted, including XPS, etc.

Answer: We do thank the Reviewer’s observation, however, we would like to point out that at present we do not see the possibility of having immediate access to analysis equipment as suggested by the Reviewer. In addition, we consider that the employed electrochemical techniques and their corresponding analyses support satisfactorily our results and contribute to the study of the use of ionic liquids as corrosion inhibitors.  

Round 2

Reviewer 2 Report

The authors have answered the questions satisfactorily.

The manuscript can be published in its current form.